# VEAttack: Downstream-agnostic Vision Encoder Attack against Large Vision Language Models

**Hefei Mei[1], Zirui Wang[1], Shen You[1], Minjing Dong[*1], Chang Xu[2]**
[1]City University of Hong Kong    [2]University of Sydney

## Abstract

Large Vision-Language Models (LVLMs) have demonstrated capabilities in multimodal understanding, yet their vulnerability to adversarial attacks raises significant concerns. To achieve practical attacking, this paper aims at efficient and transferable untargeted attacks under limited perturbation sizes. Considering this objective, white-box attacks require full-model gradients and task-specific labels, making costs scale with tasks, while black-box attacks rely on proxy models, typically requiring large perturbation sizes and elaborate transfer strategies. Given the centrality and widespread reuse of the vision encoder in LVLMs, we adopt a gray-box setting that targets the vision encoder alone for efficient but effective attacking. We theoretically establish the feasibility of vision-encoder-only attacks, laying the foundation for our gray-box setting. Based on this analysis, we propose perturbing patch tokens rather than the class token, informed by both theoretical and empirical insights. We generate adversarial examples by minimizing the cosine similarity between clean and perturbed visual features, without accessing the subsequent models, tasks, or labels. This significantly reduces computational overhead while eliminating the task and label dependence. VEAttack has achieved a performance degradation of $94.5\%$ on image caption task and $75.7\%$ on visual question answering task. We also reveal some key observations to provide insights into LVLM attack/defense: 1) hidden layer variations of LLM, 2) token attention differential, 3) Möbius band in transfer attack, 4) low sensitivity to attack steps. The code is available at https://github.com/hefeimei06/VEAttack-LVLM.

## 1 Introduction

Large Vision-Language Models (LVLMs) (Liu et al., 2023; Zhu et al., 2023; Bai et al., 2023; Team et al., 2023) have revolutionized multimodal AI by unifying pre-trained vision encoders (Radford et al., 2021; Li et al., 2022) with large language models (LLMs) (Touvron et al., 2023; Achiam et al., 2023), enabling seamless integration of visual and textual understanding for tasks including visual question answering, image captioning, *etc*. However, their reliance on vision inputs inherits adversarial vulnerabilities well studied in computer vision (Madry et al., 2018; Szegedy et al., 2014), where imperceptible perturbations can mislead model predictions. As LVLMs increasingly deploy in real-world systems, their robustness against such attacks becomes critical, especially since adversarial perturbations to vision inputs can propagate through cross-modal alignment, causing catastrophic failures in downstream text generation (Zhao et al., 2023; Cui et al., 2024; Li et al., 2025).

We explore the vulnerability of LVLMs under untargeted attacks with small perturbation sizes, which follows the standard attack setting (Madry et al., 2018). However, since LVLM targets different tasks with high computational cost, we aim at efficient attacks that can transfer across models and tasks to ensure practical attacks, which existing attacks can hardly achieve. Specifically, white-box attacks (Schlarmann & Hein, 2023; Bhagwatkar et al., 2024) normally optimize perturbations for a single task with specific labels (*e.g.*, captioning), making the cost proportional to the number of tasks. Although transferring adversarial examples to other tasks could be a naive solution, the nature of white-box attacks, *i.e.*, task and label specificity, makes them ineffective when transferred

---

*Corresponding Author

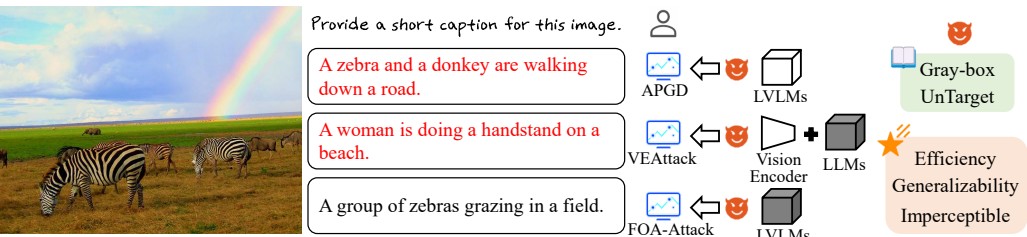

Figure 1: The illustration of different attack paradigms where the white modules are accessible to the attacker, while dark modules are inaccessible during the attack with $2/255$ as perturbation budgets.

to other tasks (*e.g.*, VQA) according to our empirical evidence in Fig. 2. Black-box attacks (Zhang et al., 2024; Jia et al., 2025) typically manipulate LVLMs to output specified targets with proxy known vision encoders (*e.g.*, CLIP-ViT-B/16) and substantial perturbation budgets. As shown in Fig. 10 and Fig.2 (b), when confronted with minimal perturbations, these methods often struggle to achieve effective attacks. One potential solution is the gray-box setting in Fig. 1, where partial model parameters are accessible, which reduces the attacking cost compared to white-box while enhancing attacking capability compared to black-box. However, it is difficult for existing gray-box attacks (Wang et al., 2024c) to achieve better satisfactory performance, even with text modality information and additional text encoders. Drawing from traditional vision tasks, where strong vision backbones boost downstream performance (Lin et al., 2017; Liu et al., 2021), and the pivotal role of vision encoders in LVLMs (Jain et al., 2024; Tong et al., 2024), we argue that the attacking capability of the vision encoder has not been fully explored and propose to investigate the feasibility of leveraging the vision encoder solely to achieve efficient and transferable attacks on LVLMs.

In this paper, we attribute this objective to explore the attack potential of the vision encoder and ensure the effectiveness of perturbation propagation to downstream LLMs, so that vision-encoder-only attack could improve generalization across multiple tasks and decrease computational costs, while maintaining effective adversarial perturbations. Specifically, we theoretically analyze the lower bound of perturbation in the multimodal aligned features for LLMs when perturbing solely on vision encoders, which provides a feasibility foundation for our attack context and gray-box attacks. Through proposing naive solutions to our question drawn from black-box attack (Zhao et al., 2023) and adversarial finetuning work (Schlarmann et al., 2024), our theoretical findings show that such perturbations have a reduced impact on class token features compared to directly targeting patch token features. Thus, we optimize perturbations of vision encoders by minimizing the cosine similarity between clean and perturbed patch token features for stronger attack effects. Through focusing merely on the vision encoder, VEAttack reduces the parameter space for attack, achieving an 8-fold reduction in time costs compared to ensemble attack (Schlarmann & Hein, 2023). It also enables downstream-agnostic generalization across diverse tasks, achieving a performance degradation of 94.5% on the image caption task and 75.7% on the visual question answering (VQA) task.

Furthermore, drawing upon extensive experiments and empirical analyses employing VEAttack, we reveal four observations to provide insights into LVLM attack or defense: 1) hidden layer variations of LLM, 2) token attention differential, 3) Möbius band in transfer attack, 4) low sensitivity to attack steps. For example, VEAttack reveals strong cross-model transferability, where perturbations crafted for one vision encoder (*e.g.*, CLIP (Radford et al., 2021)) compromise others (*e.g.*, Qwen-VL (Bai et al., 2023)) with a Möbius band phenomenon: enhancing vision encoder robustness bolsters LVLM robustness, yet paradoxically, adversarial samples targeting such robust encoders exhibit greater attack transferability across diverse LVLMs. The insights from VEAttack will inspire the research community to deepen exploration of LVLM vulnerabilities, fostering advancements in both attack and defense mechanisms to enhance the robustness and security of multimodal AI systems.

## 2 RELATED WORK

**Adversarial robustness of LVLMs.** Adversarial training (AT) (Rice et al., 2020; Dong & Xu, 2023; Lin et al., 2024; Mei et al., 2025) has proven effective in enhancing model robustness against adversarial attacks, typically in image classification tasks. With the advent of vision-language models like CLIP (Radford et al., 2021), research has shifted toward ensuring robustness and generalization in

these multimodal systems (Mao et al., 2023; Wang et al., 2024b). FARE (Schlarmann et al., 2024) improves the robustness of downstream LVLMs through an unsupervised adversarial fine-tuning scheme on the vision encoder. Recent studies (Hossain & Imteaj, 2024; Malik et al., 2025) further advance LVLM robustness by optimizing adversarial fine-tuning strategies and scaling up the vision encoder. These works reveal the critical role of a robust vision encoder in LVLM security.

**Adversarial attacks on LVLMs.** Adversarial attacks on LVLMs are normally divided into white-box, black-box, and gray-box attacks. White-box attacks (Schlarmann & Hein, 2023; Cui et al., 2024) can adapt traditional attack methods (Madry et al., 2018; Croce & Hein, 2020) to effectively target LVLMs, achieving high attack success rates. However, their task-specific strategies could lead to low generalization for multi-task system LVLMs. CroPA (Luo et al., 2024) proposes learnable prompts to effectively mislead VLMs across diverse textual inputs, but requires calculating gradients through the massive LLM. Black-box attacks (Zhao et al., 2023; Zhang et al., 2024; Li et al., 2025; Jia et al., 2025; Yang et al., 2025) leverage self-supervised learning, diffusion models, or other transfer strategies to craft effective attacks with relatively larger perturbations. Doubly-UAP (Kim et al., 2024) introduced universal perturbations trained on large-scale datasets to achieve cross-image transferability with significant pre-training costs. Gray-box attacks (Wang et al., 2024c;a; Zhang et al., 2025; Wang et al., 2023) relieve the attack objective to the visual features in LVLMs, but still introduce text information and the additional text encoder for better performance. Different from the gray-box context above, our work aims to explore using only the vision encoder and image modality to achieve a more effective and efficient attack.

## 3 PRELIMINARY

An LVLM generally comprises a pre-trained vision encoder (e.g., CLIP (Radford et al., 2021)), a large language model (LLM) (Brown et al., 2020; Touvron et al., 2023; Achiam et al., 2023), and a cross-modal alignment mechanism to achieve joint understanding of images and text. As for the vision encoder CLIP $f_{CLIP}$, it typically generates two outputs, which are class token embedding $z_{cls} \in \mathbb{R}^{1 \times d_{cls}}$ and patch tokens $z_v \in \mathbb{R}^{n_v \times d_v}$. Formally, LVLMs process an image-text pair $((v, t) \in \mathcal{V} \times \mathcal{T})$ to generate outputs conditioned on multimodal representations, where $\mathcal{V}$ represents the visual input space and $\mathcal{T}$ denotes the textual input space. In the process of prediction by LVLMs, we denote the vision encoder as $f_V$, which follows the relationship as $f_{CLIP} = \{z_{cls}; f_V(v)\}$, $f_V(v) = z_v \in \mathbb{R}^{n_v \times d_v}$, where $n_v$ is the number of patch tokens and $d_v$ is the dimension of each token. Then, the cross-modal alignment mechanism $f_A : \mathbb{R}^{n_v \times d_v} \to \mathbb{R}^{n_v \times d_m}$ aligns visual features $z_v$ with the input space of the language model, resulting in $z_m$. This feature will be combined with the textual tokens $z_t \in \mathbb{R}^{n_t \times d_t}$ from the tokenizer to form a unified input sequence. Finally, the LLM processes the input autoregressively to generate an output sequence, modeled as:

$$p(y|z_m; z_t; \theta) = \prod_{i=1}^{N} p(y_i|y_{<i}, z_m; z_t; \theta), \tag{1}$$

where $i$ is the index of the current token position, $z_m = f_A(z_v) = f_A(f_V(v))$, $z_t = tokenizer(t)$. In the traditional white-box adversarial attack (Croce & Hein, 2020), the objective is to maximize the cross-entropy loss of the model on the adversarial input, which can be formalized as:

$$\max_{\|\delta\| \leq \epsilon} \mathcal{L}(v + \delta, t; \theta) = \max_{\|\delta\| \leq \epsilon} -\log p(y \mid f_A(f_V(v + \delta)); tokenizer(t); \theta), \tag{2}$$

where $\delta$ denotes a small perturbation and $\epsilon$ is the perturbation budget.

## 4 METHODOLOGY

In this section, we explore the gray-box attack on LVLMs in a vision-encoder-only context. Specifically, we first analyze the challenges associated with existing white-box and black-box attacks under our gray-box settings, while ensuring a lower-bound perturbation for aligned features in LVLMs as a feasibility basis. With a theoretical analysis of naive solutions within our framework, we redirect the attack target and achieve an effective VEAttack.

### 4.1 REDEFINE ADVERSARIAL OBJECTIVE

To clarify our redefined adversarial objective, which shifts from full access attacks on LVLMs to targeting the vision encoder, we first establish definitions for different attacks. In traditional white-box

attacks, attackers have full access to the LVLM's architecture, including input image $v$, corresponding textual construction $t$, and model weights $\theta = \{\theta_V, \theta_A, \theta_{LLM}\}$, where the items in $\theta$ represent model weights for vision encoder $f_{CLIP}$, multimodal alignment mechanism $f_A$ and LLM $f_{LLM}$. In our VEAttack, we relieve the reliance on the white-box setting, which transforms the full model attack into a vision encoder $f_{CLIP}$, where only model weights $\theta_V$ and input images $v$ are accessible. For an image $v$ with multi-task $T = \{T^1, \cdots, T^m\}$, the corresponding constructions can be $t^T = \{t^{T^1}, \cdots, t^{T^m}\}$. Following Eq. (2), the objective of the traditional white-box for multi-task can be formulated as:

$$\max_{\|\delta^{T^i}\| \leq \epsilon} \mathcal{L}\left(\tilde{v}^{T^i}, t^{T^i}; \theta\right) = \max_{\|\delta^{T^i}\| \leq \epsilon} -\log p\left(y^{T^i} \mid f_A\left(f_V(\tilde{v}^{T^i})\right); tokenizer(t^{T^i}); \theta\right), \quad (3)$$

where $\tilde{v} = v + \delta$ is the adversarial image, $i = \{1, \cdots, m\}$ denotes the task index. It can be seen that for each task $T^i$, the traditional white-box attacks need to generate a corresponding adversarial sample $\tilde{v}^{T^i}$. In the context of LVLMs, which are designed for a wide range of downstream tasks, this task-specific framework could be ineffective when confronted with task transfer.

Although white-box attacks demonstrate potential adversarial capabilities (Schlarmann & Hein, 2023; Cui et al., 2024) on a specific task $T^i$, it's hard to maintain strong adversarial effectiveness when transferring attacks across different tasks. Black-box attacks (Li et al., 2025; Jia et al., 2025) could effectively enhance the generalization but need time-consuming transfer strategies and larger perturbations as shown in Fig. 10. In Fig. 2 (b), we conduct a transfer attack on two tasks $\{T^1, T^2\}$, which are image captioning and visual question answering with perturbation budget $\epsilon = 4/255$. For white-box APGD (Croce & Hein, 2020) and black-box FOA-Attack (Jia et al., 2025), the performance degradation of adversarial sample $\tilde{v}^{T^i}$ on task $T^j$ is not obvious, where $i, j \in \{1, 2\}$, $i \neq j$. Even in task-specific scenarios, white-box and black-box attacks typically entail higher computational costs as shown in Fig. 2 (a). Facing the above challenges, we propose targeting the vision encoder to craft ad-

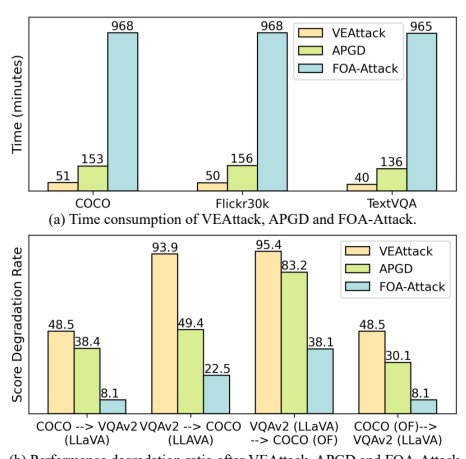

(a) Time consumption of VEAttack, APGD and FOA-Attack.

(b) Performance degradation ratio after VEAttack, APGD and FOA-Attack.

Figure 2: Transfer capability and time consumption of white-box APGD, black-box FOA-Attack, and our gray-box VEAttack.

versarial perturbations $\tilde{v}$ that exploit its shared representations across tasks $T^i$, leveraging its critical role in LVLMs (Shen et al., 2021; Tong et al., 2024) and insights from vision backbones in traditional tasks (Lin et al., 2017; Liu et al., 2021).

**Proposition 1** *For LLaVa (Liu et al., 2023) with a linear alignment layer, let $\Delta z_v = \tilde{z}_v - z_v$ denote the difference between the patch tokens output by the vision encoder CLIP before and after the perturbation, $\|\Delta z_v\|_F \geq \Delta$, $W_a$ is the weight of projection layer, $\sigma_{min}$ denotes the minimum singular value of $W_a$. Assume $\sigma_{min}(W_a) > 0$, then the difference between aligned features $z_m = f_A(z_v)$ and $\tilde{z}_m = f_A(\tilde{z}_v)$ for downstream LLMs will satisfy $\|\Delta z_m\|_F \geq \sigma_{min}(W_a)\Delta > 0$.*

*Proof.* See Appendix B.

The Proposition 1 gives a lower bound of perturbations on the aligned feature $z_m$ when attacking on $z_v$ of the vision encoder, so the perturbations from the VEAttack on $z_v$ can reliably influence downstream $f_{LLM}$, ensuring the feasibility of VEAttack. When attacking the vision encoder of LVLMs, the perturbed feature $\tilde{z}_v$ will be represented as a fundamental feature to affect any downstream tasks and LLMs. In Fig. 3, we empirically show that perturbations in the features $\tilde{z}_v$ propagate to the aligned features $\tilde{z}_m$, with increasing perturbation amplitude amplifying the feature differences in both $\Delta z_v$ and $\Delta z_m$. Following the importance of

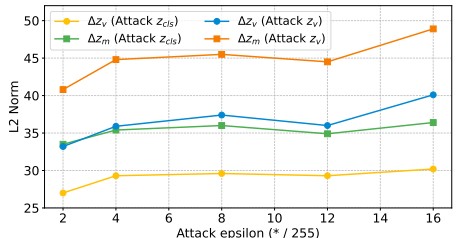

Figure 3: The feature difference before and after VEAttack with different budgets.

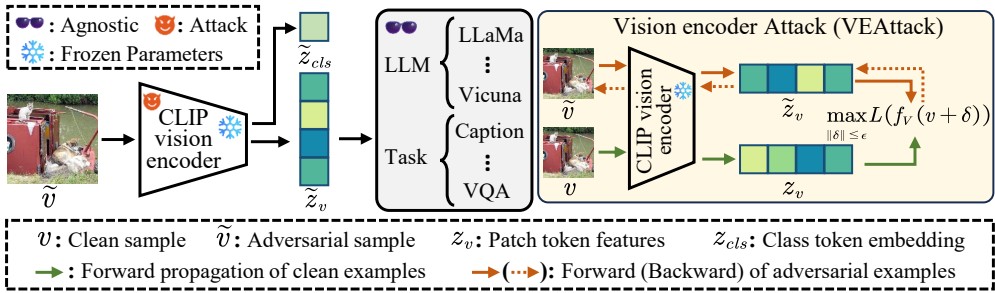

Figure 4: The illustration of the overall framework of our attack paradigm, where we solely attack the vision encoder of LVLMs within a downstream-agnostic context. The module with a yellow background is the vision encoder attack (VEAttack) method against LVLMs.

the vision encoder and theoretical analysis of the propagability of perturbations, we redefine the adversarial objective to the vision encoder and reformulate the attack objective in Eq. (3) as follows:

$$\max_{\|\delta^{T^i}\| \leq \epsilon} \mathcal{L}\left(v^{T^i} + \delta^{T^i}, t^{T^i}; \theta\right) \longrightarrow \max_{\|\delta\| \leq \epsilon} \mathcal{L}(v + \delta; \theta_V) = \max_{\|\delta\| \leq \epsilon} \mathcal{L}(f_{CLIP}(v + \delta)), \quad (4)$$

where $\theta_V$ is the weight of vision encoder, $f_{CLIP} = \{z_{cls}; f_V(v)\}$. Through Eq. (4), it can be seen that task-specific adversarial sample $\tilde{v}^{T^i}$ in all tasks $T$ could be represented by an adversarial sample $\tilde{v} = v + \delta$ in our VEAttack. Based on the redefined objective, we need to explore a suitable optimization function $\mathcal{L}(f_{CLIP}(\cdot))$ for VEAttack.

## 4.2 VISION ENCODER ATTACK

Given the VEAttack framework in Section 4.1, we can treat the attacks in adversarial training algorithms proposed by robust CLIP work (Schlarmann et al., 2024) and black-box attack (Zhao et al., 2023) as naive solutions to tackle Eq. (4). These vanilla attacks can be formulated as:

$$\text{AttackVLM-ii}: \ \tilde{v} = \arg\max_{\|\delta\| \leq \epsilon} -\cos\left(\tilde{z}_{cls}, z_{cls}\right) \quad L_2 \text{ Attack}: \ \tilde{v} = \arg\max_{\|\delta\| \leq \epsilon} \|\tilde{z}_{cls} - z_{cls}\|_2^2, \quad (5)$$

The proposed naive solutions are all classification-centric and target at class token embedding $z_{cls}$ within the CLIP vision encoder $f_{CLIP} = \{z_{cls}; f_V(v)\}$. Unlike classification tasks, LVLMs typically rely on the patch token features $z_v = f_V(v)$ during inference, so the naive attacks need to result in a perturbed image $\tilde{v}$ first and then indirectly affect the visual feature $\tilde{z}_v$. This indirect perturbation may limit the effectiveness of attacks on LVLMs, which we will analyze theoretically.

**Proposition 2** *Consider two kinds of attack targets: 1) If the perturbation is introduced to the single class token $z_{cls}$ and propagates through $z_{cls} \xrightarrow{backward} v \xrightarrow{forward} z_v$. Let $\Delta z_{cls}(z_{cls})$, $\Delta z_v(z_{cls})$, $\Delta z_m(z_{cls})$ denote the perturbations on features $z_{cls}$, $z_v$ and $z_m$ through the first attack, respectively. 2) If the perturbation is directly introduced to the patch token features $z_v$, let $\Delta z_m(z_v)$ denote the perturbation on the aligned feature $z_m$ through the second attack. Assume the same degree of perturbation on $z_{cls}$ and $z_v$ during the two attacks, then the ratio of the effect is given by:*

$$\frac{\|\Delta z_m(z_{cls})\|_F}{\|\Delta z_m(z_v)\|_F} = \frac{\|\Delta z_v(z_{cls})\|_F}{\|\Delta z_{cls}(z_{cls})\|_2} \leq \frac{3 + \epsilon_V}{\sqrt{n_v}}, \ \epsilon_V \ll 1. \quad (6)$$

*Proof.* See Appendix C.

From Proposition 2, we infer that perturbations on the class token $z_{cls}$ have a reduced impact on the patch token features $z_v$, as the ratio $(3 + \epsilon_V)/\sqrt{n_v}$ is typically smaller than 1 (*e.g.*, $\sqrt{n_v} = 16$ in CLIP-ViT-L/14). Consequently, this diminished effect propagates to the aligned features $z_m$, weakening the adversarial impact on downstream tasks compared to directly targeting $z_v$. In Fig. 3, we empirically demonstrate that under various perturbation budgets, the perturbations $\|\Delta z_v\|_F$ and $\|\Delta z_m\|_F$ induced by attacking $z_{cls}$ are significantly smaller than those from directly attacking $z_v$, further validating the superior effectiveness of targeting $z_v$. Therefore, we reformulate the objective

function from Eq. (4) to directly target $z_v$, expressed as $\max_{\|\delta\|\leq\epsilon} \mathcal{L}(f_V(v+\delta))$. As illustrated in Fig. 4, VEAttack targets the patch token features $z_v$ of the vision encoder in a downstream-agnostic context. We adopt cosine similarity as the loss metric for $f_V(\cdot)$, as $\ell_2$ loss may concentrate perturbations on specific dimensions, with limited impact on $z_m$ if the alignment layer is less sensitive to those dimensions. In contrast, cosine similarity holistically perturbs the semantic direction of features $z_v$, effectively propagating to $z_m$. Thus, with cosine similarity as the optimization metric, the generated examples in our VEAttack can be formulated as:

$$\tilde{v} = \arg\max_{\|\delta\|\leq\epsilon} -\cos\left(f_V(v+\delta), f_V(v)\right) = \arg\max_{\|\delta\|\leq\epsilon} -\cos(z_v+\delta, z_v). \tag{7}$$

## 5 EXPERIMENTS

### 5.1 EXPERIMENTAL SETTINGS

**Gray-box attack setup.** We evaluate the attack performance of our method and baseline methods on pre-trained LVLMs and conduct PGD attack (Madry et al., 2018) on the vision encoder CLIP-ViT-L/14[1]. Following the white-box attack settings (Schlarmann et al., 2024), the perturbation budgets are set as $\epsilon = 2/255$ and $\epsilon = 4/255$, attack stepsize $\alpha = 1/255$, attack steps are $t = 100$ iterations. The attack steps of gray-box methods MIX.Attack (Tu et al., 2023) and VT-Attack (Wang et al., 2024c) are $t = 1000$. A subset of 500 randomly chosen images is used for adversarial evaluations, with all samples adopted for clean evaluations. The performance degradation ratio after the attack is utilized to assess the attack effect. All experiments were conducted on a NVIDIA-A6000 GPU.

**Pre-trained models and tasks.** Experiments are conducted on diverse LVLMs where LLaVa1.5-7B (Liu et al., 2023), LLaVa1.5-13B (Liu et al., 2023) and OpenFlamingo-9B (OF-9B) (Awadalla et al., 2023) are evaluated against VEAttack. In addition to them, we adopt miniGPT4 (Zhu et al., 2023), mPLUG-Owl2 (Ye et al., 2024), and Qwen-VL (Bai et al., 2023) for transfer attack of VEAttack. For the image caption task, we evaluate the performance on COCO (Lin et al., 2014) and Flickr30k (Plummer et al., 2015) datasets. For the VQA task, our experiments are conducted on TextVQA (Singh et al., 2019) and VQAv2 (Goyal et al., 2017). All image caption tasks are evaluated using CIDEr (Vedantam et al., 2015) score, while VQA tasks are measured by VQA accuracy (Antol et al., 2015). Beyond these tasks, we additionally evaluate our VEAttack on a hallucination benchmark called POPE (Li et al., 2023b), and its evaluation metric is F1-score.

**Transfer attack setting.** The source models contain variants of the CLIP-ViT-L/14 model, which are pre-trained from OpenAI[1], TeCoA (Mao et al., 2023) and FARE (Schlarmann et al., 2024). Both TeCoA and FARE employ adversarial training with a perturbation budget of $\epsilon = 4/255$. The target models encompass these three CLIP variants as well as LVLMs from the pre-trained model settings, which utilize vision encoders other than CLIP. Following the setting of transfer attack (Wei et al., 2022; Chen et al., 2023), we assess transfer attacks across different CLIP models with $\epsilon = 8/255$ and transfer attacks between CLIP and non-CLIP vision encoders with $\epsilon = 16/255, \alpha = 16/(255 \times 5)$. All transfer attacks are conducted with $t = 100$ iteration steps.

### 5.2 VEATTACK ON LVLMS

We compare VEAttack with gray-box attacks in Table 1 across diverse LVLMs and tasks. For the image caption task, VEAttack demonstrates a highly effective attack, reducing the average CIDEr score on COCO by $94.5\%$ under a perturbation of $\epsilon = 4/255$. In the VQA task, our proposed VEAttack method also exhibits competitive performance, reducing the average accuracy on the TextVQA dataset by $75.7\%$ under a perturbation of $\epsilon = 4/255$. Conversely, other attacks exhibit lower attack effectiveness compared to our method in the vision encoder setting. On the POPE with F1-score metric, values below $50\%$ on balanced datasets indicate poor performance. VEAttack significantly degrades performance under a perturbation of $\epsilon = 4/255$, achieving a $38.0\%$ reduction in F1-score and driving it below $50\%$, indicating the VEAttack could increase hallucination in LVLMs.

**Observation 1** *Even though the LLMs and tasks are downstream-agnostic, an attack on the output of the vision encoder can lead to variations in the hidden layer features of the LLM.*

---

[1]https://github.com/openai/CLIP

Table 1: Comparison of VEAttack with different gary-box attacks across different LVLMs and tasks.

| Task | LVLMs / Attack | OpenFlamingo-9B $\epsilon = 2/255$ | $\epsilon = 4/255$ | LLaVa1.5-7B $\epsilon = 2/255$ | $\epsilon = 4/255$ | LLaVa1.5-13B $\epsilon = 2/255$ | $\epsilon = 4/255$ | Average $\epsilon = 2/255$ | $\epsilon = 4/255$ |
|---|---|---|---|---|---|---|---|---|---|
| COCO | Clean | 79.7 | | 115.5 | | 119.2 | | 104.8(↓0.0%) | |
| | MIX.Attack Tu et al. (2023) | 45.9 | 25.4 | 67.5 | 55.4 | 73.8 | 60.1 | 62.4(↓40.5) | 47.0(↓55.1) |
| | VT-Attack Wang et al. (2024c) | 38.9 | 21.6 | 50.8 | 12.2 | 58.2 | 20.1 | 49.3(↓52.9) | 18.0(↓82.8) |
| | AttackVLM-ii Zhao et al. (2023) | 24.4 | 10.3 | 40.9 | 25.8 | 42.7 | 27.5 | 36.0(↓65.6) | 21.2(↓79.8) |
| | VEAttack | **7.5** | **3.7** | **10.8** | **7.1** | **11.2** | **6.5** | **9.8(↓90.6)** | **5.8(↓94.5)** |
| Flickr30k | Clean | 60.1 | | 77.5 | | 77.1 | | 71.6 | |
| | MIX.Attack Tu et al. (2023) | 33.7 | 18.0 | 42.1 | 35.9 | 41.0 | 32.2 | 38.9(↓45.7) | 28.7(↓59.9) |
| | VT-Attack Wang et al. (2024c) | 27.7 | 13.8 | 35.1 | 12.3 | 34.0 | 14.6 | 32.3(↓54.9) | 13.6(↓81.0) |
| | AttackVLM-ii Zhao et al. (2023) | 18.1 | 9.9 | 29.9 | 19.8 | 29.7 | 21.6 | 25.9(↓63.8) | 17.1(↓76.1) |
| | VEAttack | **8.7** | **3.2** | **10.7** | **6.3** | **9.1** | **5.7** | **9.5(↓86.7)** | **5.1(↓92.9)** |
| TextVQA | Clean | 23.8 | | 37.1 | | 39.0 | | 33.3 | |
| | MIX.Attack Tu et al. (2023) | 13.4 | 8.8 | 24.6 | 19.1 | 22.8 | 19.7 | 20.3(↓39.0) | 15.9(↓52.3) |
| | VT-Attack Wang et al. (2024c) | 15.3 | 10.5 | 23.7 | **10.0** | 24.9 | 10.0 | 21.3(↓36.0) | 10.2(↓69.4) |
| | AttackVLM-ii Zhao et al. (2023) | **12.1** | 7.6 | 19.7 | 11.9 | 19.8 | 14.2 | 17.2(↓48.3) | 11.2(↓66.4) |
| | VEAttack | 12.5 | **5.7** | **13.8** | 10.1 | **12.4** | **8.6** | **12.9(↓61.3)** | **8.1(↓75.7)** |
| VQAv2 | Clean | 48.5 | | 74.5 | | 75.6 | | 66.2 | |
| | MIX.Attack Tu et al. (2023) | 39.8 | 36.0 | 59.4 | 57.9 | 59.8 | 58.0 | 53.0(↓19.9) | 50.6(↓23.6) |
| | VT-Attack Wang et al. (2024c) | 38.5 | 37.0 | 53.6 | **21.4** | 55.2 | **21.0** | 49.1(↓25.8) | **26.5(↓59.9)** |
| | AttackVLM-ii Zhao et al. (2023) | 37.5 | 35.8 | 54.1 | 49.7 | 56.2 | 49.6 | 49.3(↓25.5) | 45.0(↓32.0) |
| | VEAttack | **34.0** | **32.8** | **42.9** | 38.4 | **41.5** | 37.6 | **39.5(↓40.3)** | 36.3(↓45.2) |
| POPE | Clean | 65.7 | | 84.5 | | 84.4 | | 78.1 | |
| | MIX.Attack Tu et al. (2023) | 59.0 | 53.3 | 72.0 | 69.1 | 74.2 | 68.2 | 68.4(↓12.4) | 63.5(↓18.7) |
| | VT-Attack Wang et al. (2024c) | 63.6 | 63.5 | 64.0 | 60.1 | 65.5 | 66.4 | 64.4(↓17.5) | 63.3(↓18.9) |
| | AttackVLM-ii Zhao et al. (2023) | 53.3 | 48.1 | 69.0 | 61.6 | 63.8 | 57.8 | 49.3(↓36.9) | 45.0(↓42.4) |
| | VEAttack | **60.6** | **59.6** | **47.5** | **42.8** | **47.6** | **44.7** | **51.9(↓33.5)** | **49.0(↓37.3)** |

We visualize the first visual token features of an LLM's hidden layers using t-SNE in Fig. 5 for clean and adversarial samples. To more effectively demonstrate the relative positions of the features in each layer, we denote the positions of the 0th, 16th, and 32nd features. For clean inputs in (a), features show distinct clustering across layers, with progressive spread from Layer 0 to 32. For adversarial inputs in (b), features scatter widely, and the relative relationships between features are more biased.

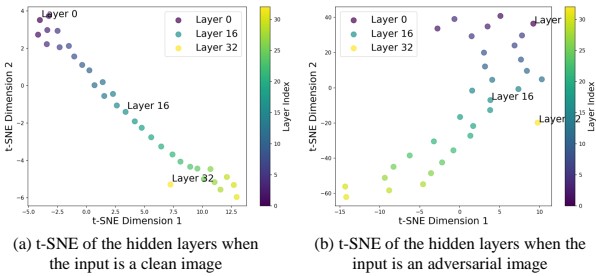

(a) t-SNE of the hidden layers when the input is a clean image

(b) t-SNE of the hidden layers when the input is an adversarial image

Figure 5: t-SNE visualization of the features across hidden layers in LLM for clean and adversarial inputs.

*This confirms that attacking the vision encoder induces notable variations in the LLM's hidden layer representations, despite being in a downstream-agnostic context, validating the effectiveness of our vision-encoder-only attack.*

**Observation 2** *LVLMs will pay more attention to image tokens in the image caption task. Conversely, LVLMs focus more on user instruction tokens in the VQA task.*

As shown in Table 1, following an identical attack on the vision encoder, the performance degradation in the VQA task is slightly less than that observed in the image caption task. Consequently, we speculate that the VQA task may not pay as much attention to the image tokens as the image caption task. To this end, Fig. 6 shows the attention map between the output tokens and the system tokens, image tokens, and instruction tokens within the shallow Layer1, the middle Layer16, and the output Layer32. In the COCO caption task, we observe higher attention scores between output tokens and image tokens, with an average attention value of 0.10 compared to 0.08 in the VQA task. Conversely, in the VQA task, output tokens exhibit stronger attention to instruction tokens, with an average attention value of 0.47, compared to 0.35 in the COCO caption task. This supports our hypothesis that instruction tokens play a more critical role in VQA, while image tokens are more pivotal for image captioning in LVLMs. *We hope this observation of differential reliance on image*

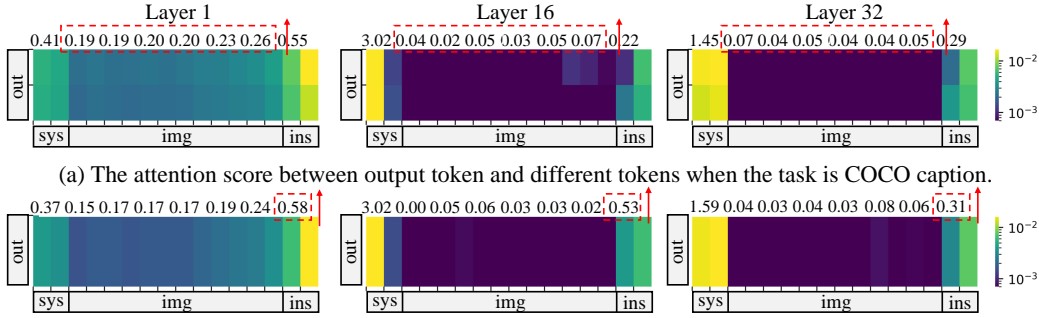

(a) The attention score between output token and different tokens when the task is COCO caption.

(b) The attention score between output token and different tokens when the task is VQAv2.

Figure 6: Illustration of attention maps across different layers of the language model for two tasks. The attention scores shown represent the relationship between the output token and three input token types (out: output tokens, sys: system tokens, img: image tokens, ins: instruction tokens).

*and instruction tokens could guide future researchers in optimizing performance of LVLMs* (Zhang et al., 2021; Xiong et al., 2024; Chen et al., 2024).

## 5.3 TRANSFER ATTACK ON LVLMS

**Observation 3** *The transfer attack on LVLMs normally resembles a Möbius band, where robustness and vulnerability of LVLMs intertwine as a single, twisted continuum. For defenders, using a more robust vision encoder can enhance the ability of LVLM to resist attacks. Conversely, for attackers, adversarial samples obtained by attacking a more robust vision encoder usually have higher attack transferability on diverse LVLMs.*

We first observe transfer attacks using CLIP with varying robustness, as shown in the diagonal performance matrix of Table 2. Employing a more robust CLIP vision encoder significantly enhances LVLM resilience against VEAttack. Specifically, FARE improves average performance by 42.9 over the CLIP of OpenAI, while TeCoA yields a 39.3 improvement. Analyzing the off-diagonal elements of the performance matrix, we observe that adversarial samples crafted against more robust vision encoders exhibit stronger transferability. For instance, attacking FARE reduces the average performance of LVLMs using CLIP as the vision encoder by 58.5%, whereas attacking CLIP exhibits limited transferability to other robust vision encoders. Extending our analysis to non-CLIP vision encoders, we find the Möbius band of VEAttack: Strengthening defenses paradoxically fuels more potent attacks, perpetuating a cyclical interplay. Notably, attacking the robust trained vision encoder FARE reduces the performance of miniGPT4 by 90.8% and the performance of mPLUG-Owl2 by 33.5%. *We hope that these findings highlight the critical need for future research to prioritize the security of vision encoders in LVLMs to mitigate adversarial vulnerabilities.*

Table 2: Transfer Attack of original CLIP and robust CLIP after adversarial training on the COCO dataset. Bold indicates the best attack performance, while underlined indicates the second best.

| Target⟍ Source | LLaVa1.5-7B | | | OpenFlamingo-9B | | | MiniGPT-4 BLIP-2 | mPLUG-Owl2 MplugOwl | Qwen-VL CLIP-bigG |
|---|---|---|---|---|---|---|---|---|---|
| | CLIP | TeCoA | FARE | CLIP | TeCoA | FARE | | | |
| clean | 115.5 | 88.3 | 102.4 | 79.7 | 66.9 | 74.1 | 96.7 | 132.3 | 138.5 |
| CLIP Radford et al. (2021) | **3.7** | 92.2 | 100.4 | **3.6** | 70.7 | 78.1 | 55.2 | 124.9 | 131.2 |
| TeCoA Mao et al. (2023) | 22.1 | **25.3** | 25.1 | 17.7 | **20.8** | 22.6 | **11.0** | 96.4 | 102.7 |
| FARE Schlarmann et al. (2024) | 38.7 | 54.5 | **48.6** | 26.3 | 42.7 | **36.4** | 8.9 | **88.0** | **101.4** |

To provide a mechanistic explanation for the "Möbius Band" effect (Observation 3), we visualize the dynamics of feature deviation during the attack process. Specifically, we track the difference between the adversarial patch token values and the clean token values across 100 attack steps. We compare three scenarios under a unified metric scale, as shown in Figure 7, which are 1) Attack CLIP, Evaluate on CLIP; 2) Attack FARE, Evaluate on CLIP; 3) Attack CLIP, Evaluate on FARE. When attacking the robust FARE model, the heatmap shows the deepest blue color, appearing early

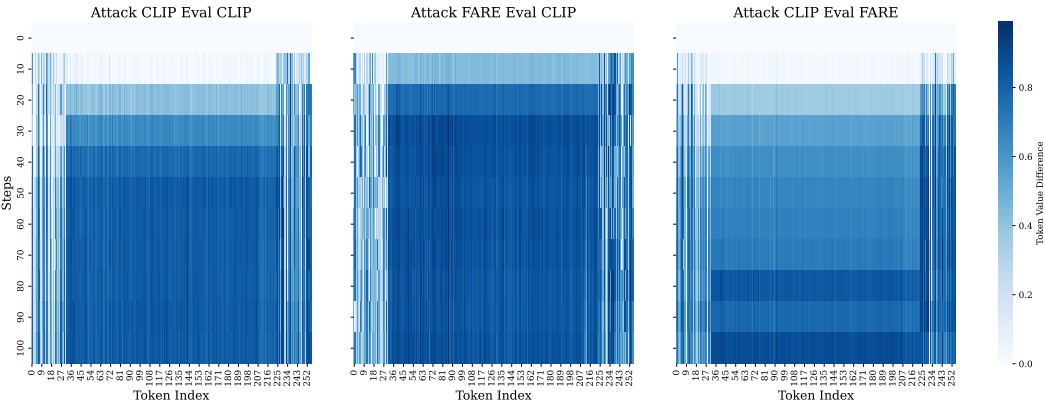

Figure 7: Feature deviation heatmaps of tokens across attack steps under different transfer settings.

in the attack steps. This indicates that because FARE is resistant to small, high-frequency noise, the optimization process is forced to induce large deviations in the semantic feature space to successfully fool it. When these strong perturbations are transferred to the standard CLIP model, they cause catastrophic feature distortion, explaining the high transfer success rate. In contrast, when attacking CLIP and transferring to FARE, the heatmap remains consistently light. This suggests that the perturbations optimized on CLIP are relatively superficial, which may sufficient to fool CLIP's decision boundary, but fail to penetrate the robustness of FARE. The robust encoder effectively filters out these specific, lower-magnitude perturbations, resulting in minimal feature deviation and attack failure. In summary, Möbius paradox shows that robust models force the attacker to learn universally destructive features, whereas standard models allow for model-specific fragile features.

## 5.4 Ablation Studys

**Observation 4** *Reducing attack steps does not significantly impair the effectiveness of VEAttack, while increasing the perturbation budget beyond a threshold within an imperceptible range does not continuously degrade the performance of LVLMs significantly.*

To further optimize the efficiency of VEAttack, we conducted ablation studies on the attack iteration $t$ and perturbation budget $\epsilon$. Despite aligning with white-box attack settings at $t = 100$, a setting of $t = 10$ still achieves a performance reduction of $84.4\%$, and $t = 50$ suffices to attain competitive performance. For the perturbation budget, performance degradation plateaus at $\epsilon = 8/255$, with further increases yielding relatively minor effects. We hypothesize this saturation stems from a gap between the output of the vision encoder $z_v$ and the aligned features $z_m$. By adopting a lower $t$ and an optimized $\epsilon$, we enhance attack efficiency, enabling complete testing on datasets, which is typically challenging in traditional white-box attacks.

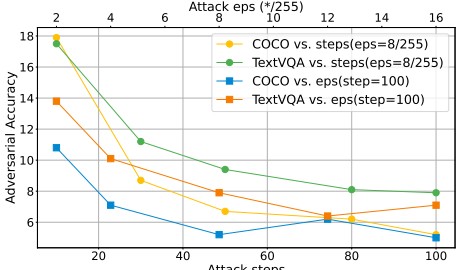

Figure 8: VEAttack performance under different steps and budgets on COCO and TextVQA datasets (top x-axis: perturbation budget, bottom x-axis: attack steps).

Following the above observation, we evaluate VEAttack with $t = 50$ and $\epsilon = 8/255$ on full datasets, comparing the performance and efficiency with white-box and gray-box attacks. The time costs for traditional white-box methods are estimated based on their evaluated subset size relative to the full dataset, and the Flops count the computation of forward once. VEAttack surpasses APGD in the image caption task and shows competitive performance in the VQA task. Crucially, compared to the comparison in Fig. 2(a), VEAttack further improved attack efficiency, reducing time costs by about 8-fold compared to white-box Ensemble attack, and about 13-fold compared to gray-box VT-Attack with 1000 iterations. *We hope this efficient implementation demonstrates the potential for practical and scalable adversarial attacks on LVLMs, providing valuable insights for future evaluations.*

Table 3: Comparison of the effectiveness and efficiency between VEAttack and other white-box and gray-box attacks. Flops count the computation of forward once.

| Attack | Version | Flops | COCO | Time (h) | Flickr30k | Time (h) | TextVQA | Time (h) |
|---|---|---|---|---|---|---|---|---|
| clean | None | 99.3G | 115.5 | 1.33 | 77.5 | 0.3 | 37.1 | 0.42 |
| APGD Croce & Hein (2020) | White-box | 9.93T | 13.1 | 25.5 | 9.5 | 5.2 | 8.1 | 22.7 |
| Ensemble Schlarmann & Hein (2023) | White-box | 9.93T | **3.1** | 41.9 | **1.2** | 14.3 | **0.0** | 37.8 |
| VT-Attack Wang et al. (2024c) | Gray-box | 3.04T | 12.2 | 65.0 | 12.3 | 13.3 | 10.0 | 64.8 |
| VEAttack ($\epsilon = 4/255, t = 100$) | Gray-box | 2.59T | 7.1 | 8.5 | 6.3 | 1.7 | 10.1 | 6.7 |
| VEAttack ($\epsilon = 8/255, t = 50$) | Gray-box | 2.59T | 5.5 | **5.3** | 4.5 | **1.1** | 9.4 | **4.1** |

**Impact of different attack targets.** Table 4 shows the results of using different tokens as attack targets on LLaVa1.5-7B, specifically class token embedding $z_{cls}$ and patch token features $z_v$. We observe that solely perturbing $z_{cls}$ results in relatively minor attack performance across all datasets, indicating its limited ability to effectively disrupt LVLMs. In contrast, simultaneously attacking both $z_{cls}$ and $z_v$ leads to more noticeable attack perfor-

Table 4: Ablation of VEAttack targets on diverse tasks with cosine similarity as the loss metric.

| Objective | | COCO ($\epsilon$)↓ | | VQAv2 ($\epsilon$)↓ | | POPE ($\epsilon$)↓ | |
|---|---|---|---|---|---|---|---|
| $z_v$ | $z_{cls}$ | $2/255$ | $4/255$ | $2/255$ | $4/255$ | $2/255$ | $4/255$ |
| ✗ | ✓ | 43.6 | 25.5 | 56.0 | 50.0 | 69.4 | 59.1 |
| ✓ | ✓ | 22.0 | 10.5 | 46.4 | 42.0 | 59.2 | 46.0 |
| ✓ | ✗ | **10.8** | **7.1** | **42.9** | **38.4** | **47.5** | **42.8** |

mance, demonstrating the superior effectiveness of $z_v$ over $z_{cls}$. Notably, the greatest attack performance is achieved when attacking only $z_v$, indicating that $z_v$ can sufficiently represent visual information. These results are consistent with our theoretical analysis in Section 4.2, confirming that perturbations on $z_v$ propagate more effectively to the downstream, resulting in effective attacks.

**Choice of loss metrics.** We compare Euclidean distance, Kullback-Leibler (K-L) divergence, and cosine similarity as the loss metrics of the objective for attacking LLaVa1.5-7B in Table 5. Experimental results show that across all benchmark datasets and perturbation budgets, using cosine similarity consistently leads to the most significant attack performance, in-

Table 5: Ablation of loss metrics in attack objective.

| Measurement | COCO ($\epsilon$)↓ | | VQAv2 ($\epsilon$)↓ | | POPE ($\epsilon$)↓ | |
|---|---|---|---|---|---|---|
| | $2/255$ | $4/255$ | $2/255$ | $4/255$ | $2/255$ | $4/255$ |
| Euclidean | 46.9 | 39.6 | 56.1 | 53.6 | 62.1 | 64.3 |
| K-L divergence | 70.0 | 34.3 | 59.4 | **33.2** | 70.0 | 60.5 |
| Cosine similarity | **10.8** | **7.1** | **42.9** | 38.4 | **47.5** | **42.8** |

dicating its superior effectiveness in adversarial attacks. This finding further validates the superior effectiveness of cosine similarity, confirming its ability to more effectively capture semantic differences between clean and adversarial samples in the feature space.

## 6 CONCLUSION

We propose VEAttack, a novel gray-box attack targeting solely the vision encoder of LVLMs, balancing reliance on task-specific gradients to overcome the task-specificity and high computational costs. By minimizing cosine similarity between clean and perturbed visual features under limited perturbation sizes, VEAttack ensures the generalization across tasks like image captioning and visual question answering, achieving performance degradation of $94.5\%$ and $75.7\%$, respectively. It reduces computational overhead 8-fold compared to white-box Ensemble attack and 13-fold compared to gray-box VT-Attack. Our experiments reveal key LVLM vulnerabilities, including hidden layer variations, token attention differentials, Möbius band in transfer attack, and low sensitivity to attack steps. We believe the insights of VEAttack will drive further research into LVLM robustness, fostering advanced defense mechanisms for secure multimodal AI systems.

**Limitations.** A limitation of this work is that we have not yet investigated defense mechanisms to counter VEAttack. Although more robust vision encoders may improve LVLM defenses, significant performance degradation persists under gray-box conditions of the vision encoder. And mitigating transfer attacks from adversarial samples targeting such robust vision encoders is unresolved. Addressing these issues is crucial for ensuring the secure deployment of LVLMs.

ACKNOWLEDGMENTS

This work was supported in part by Young Scientist Fund (No. 62406265) of NSFC, Start-up Grant (No. 9610680) of the City University of Hong Kong, and the Australian Research Council under Projects DP240101848 and FT230100549.

REPRODUCIBILITY STATEMENT

We aim to make VEAttack fully reproducible. The algorithmic specification and assumptions are given in Section 4.1 to Section 4.2) (including Eqs. (4)–(7)), with the overall pipeline summarized in Fig. 4, while formal guarantees and complete proofs appear in Appendix B and C. Experimental settings, including model variants, datasets, evaluation metrics, and the gray-box attack settings, are introduced in Section 5.1, while primary results and ablations are reported in Section 5.2 to Section 5.4 (experimental results are shown in Tables 1 to Table 5 and Figs. 2 to 8). Additional materials that aid verification include multi-task transfer results (Appendix D, Table 6), effects and costs of joint LLM (Appendix E, Table 7), extended captioning metrics (Appendix F, Table 8), Image-Text Retrieval task (Appendix G, Table 10), transfer attack from larger specific vision encoders (Appendix I, Table 13), classification evaluations (Appendix J, Table 14), and qualitative analyses (Appendix K, Figs. 10 to 13). To facilitate replication, our supplementary materials contain an anonymous code file that implements VEAttack, along with a README and run scripts that regenerate all our results.

ETHICS STATEMENT

We affirm adherence to the Code of Ethics throughout this work. Our study investigates adversarial robustness of LVLMs in a research-only setting and involves no human-subjects experiments, user studies, or collection of personal data. All experiments use publicly available datasets and pretrained models (*e.g.*, COCO, Flickr30k, TextVQA, VQAv2, LLaVA, OpenFlamingo, miniGPT-4, mPLUG-Owl2, Qwen-VL, and CLIP), following their licenses and prescribed evaluation protocols, and all sources are cited in the paper. We recognize the dual-use risk inherent in security-oriented research on adversarial attacks. We intend to expose vulnerabilities to improve robustness: we study an untargeted, vision-encoder–only attack under small perturbation budgets, report results transparently, and release any accompanying code solely to support reproducibility and robustness evaluation, not real-world misuse.

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

## A ETHICAL STATEMENT AND RESPONSIBLE DISCLOSURE

As Large Vision-Language Models (LVLMs) are increasingly deployed in safety-critical applications, investigating their adversarial vulnerabilities is essential for building resilient systems. This work introduces VEAttack, a method designed to expose inherent weaknesses in the vision encoder component of LVLMs. While our research demonstrates effective attack capabilities, our primary motivation is to advance the community's understanding of multimodal robustness and to facilitate the development of stronger defense mechanisms.

**Responsible Release.** To mitigate potential misuse, we adhere to responsible disclosure principles. We will release the source code and generated adversarial examples strictly for research purposes under a restrictive license that prohibits malicious use. Furthermore, we have consciously excluded any functionality that targets specific harmful or offensive content generation, focusing solely on benign benchmarks to demonstrate technical vulnerabilities without generating toxic outputs.

**Safety Implications and Mitigation.** The Möbius Band phenomenon identified in this study highlights a critical safety paradox: robust vision encoders can inadvertently serve as potent sources for transfer attacks. This finding underscores the urgent need for holistic defense strategies that go beyond standard adversarial training. We advocate for future research to focus on: 1) developing transfer-resistant vision encoders that decouple robustness from attack transferability, and 2) implementing multi-stage verification protocols in LVLM pipelines to detect and filter adversarial visual tokens before they propagate to the language model.

By publicly documenting these vulnerabilities and their mechanistic underpinnings, we aim to provide the necessary groundwork for the research community to preemptively secure future multimodal systems against emerging gray-box threats.

## B PROOF OF PROPOSITION 1

The cross-modal alignment mechanism $f_A : \mathbb{R}^{n_v \times d_v} \to \mathbb{R}^{n_v \times d_m}$ is a linear projection layer with weight $W_a$, where $z_v \in \mathbb{R}^{n_v \times d_v}$ is the output of vision encoder. After our visual tokens attack on vision encoder CLIP, the difference of output features during attack will have a lower bound, which can be denoted as $\Delta z_v = \tilde{z}_v - z_v$ and $\|\Delta z_v\|_F \geq \Delta$, where $\tilde{z}_v$ is the output of vision encoder after adversarial attack. So the aligned feature can be formulated as:

$$z_m = f_A(z_v) = z_v W_a, \; \tilde{z}_m = f_A(\tilde{z}_v) = \tilde{z}_v W_a, \tag{8}$$

where $z_m, \tilde{z}_m \in \mathbb{R}^{n_v \times d_m}$ is the aligned features of the clean and adversarial samples for LLM. The difference between them can be formulated as:

$$\tilde{z}_m - z_m = \tilde{z}_v W_a - z_v W_a = (\tilde{z}_v - z_v) W_a = \Delta z_v W_a \tag{9}$$

Here, we need to analyse the lower bound of $\|\Delta z_v W_a\|_F$. For the convenience of derivation, we vectorize the Frobenius norm as $\|\Delta z_v W_a\|_F = \|\text{vec}(\Delta z_v W_a)\|_2$, where $\text{vec}(\cdot)$ flattens the matrix into a vector. The vector in the above norm can be further expanded as $\text{vec}(\Delta z_v W_a) = (W_a^T \otimes I_{n_v}) \text{vec}(\Delta z_v)$, where $\otimes$ is Kronecker product, $I_{n_v}$ is a $n_v \times n_v$ identity matrix. For matrix $W_a^T \otimes I_{n_v}$, its singular values are the products of all possible singular values of $\sigma(W_a^T)$ and $\sigma(I_{n_v})$, so its smallest singular value is:

$$\sigma_{\min}(W_a^T \otimes I_{n_v}) = \sigma_{\min}\left(W_a^T\right) \cdot \sigma_{\min}\left(I_{n_v}\right) = \sigma_{\min}\left(W_a\right) \cdot 1 = \sigma_{\min}\left(W_a\right) \tag{10}$$

For any matrix $A$ and vector $x$, The $\ell_2$ norm transformation satisfies $\|Ax\|_2 \geq \sigma_{\min}(A)\|x\|_2$. In our derivation, $x = \text{vec}(\Delta z_v)$, $A = W_a^T \otimes I_{n_v}$, so we can get the following inequality relationship:

$$\left\|W_a^T \otimes I_{n_v} \text{vec}\left(\Delta z_v\right)\right\|_2 \geq \sigma_{\min}(W_a^T \otimes I_{n_v}) \left\|\text{vec}\left(\Delta z_v\right)\right\|_2 = \sigma_{\min}\left(W_a\right) \left\|\Delta z_v\right\|_F \tag{11}$$

As $\|\Delta z_v W_a\|_F = \|\text{vec}(\Delta z_v W_a)\|_2 = \left\|W_a^T \otimes I_{n_v} \text{vec}\left(\Delta z_v\right)\right\|_2$, we can get:

$$\|\tilde{z}_m - z_m\|_F = \|\Delta z_v W_a\|_F \geq \sigma_{\min}\left(W_a\right) \|\Delta z_v\|_F \geq \sigma_{\min}\left(W_a\right) \Delta \tag{12}$$

So we can get the lower bound of aligned features as $\sigma_{\min}\left(W_a\right) \Delta$. In the context of the LLaVA model, the alignment layer $f_A$ maps the output of the visual encoder (e.g., CLIP ViT-L/14, with dimension $d_v = 1024$) to the language model embedding space (e.g., Vicuna, with dimension $d_m =$

4096), where the weight matrix $W_a \in \mathbb{R}^{d_v \times d_m}$ satisfies $d_m > d_v$. In LLaVA, the projection layer is typically trained to align visual and language features effectively, which encourages $W_a$ to achieve full rank (i.e., $\text{rank}(W_a) = d_v$) to maximize the expressiveness of the mapping. Empirical experiments on the LLaVa1.5-7B model reveal that the minimum singular value of the projection layer is approximately 0.597, corroborating the theoretical expectation of a positive lower bound. Under this condition, $\sigma_{\min}(W_a) > 0$, ensuring the existence of the lower bound $\sigma_{\min}(W_a) \cdot \Delta > 0$.

## C   PROOF OF PROPOSITION 2

The vision encoder CLIP-ViT processes an input image by dividing it into $n_v$ patch tokens $z_v$ and a class embedding $z_{cls}$. In Layer 0, the initial token can be denoted as $T_0 = [z_{cls}^0; z_v^{0,1}; \ldots; z_v^{0,n_v}] \in \mathbb{R}^{(n_v+1) \times d_v}$, where $z_v^{0,i}$ represents the embedding of the $i$-th patch. The ViT comprises $L$ layers, applying a self-attention mechanism followed by a residual connection:

$$T_{l+1} = \text{Attention}(T_l) + T_l, \quad l = 0, \ldots, L-1,$$

where the attention operation is defined as:

$$\text{Attention}(T_l) = A_{d_v \times d_v} V = \text{softmax}\left(\frac{QK^T}{\sqrt{d}}\right) V,$$

with $Q = T_l W_Q$, $K = T_l W_K$, and $V = T_l W_V$ where the three weights are learned projection matrices. The final layer output $T_L = [z_{cls}^L; z_v^L]$ provides the token representations, where $z_v^L$ is aligned to the language model input via $z_m = W_a z_v^L$, $W_a \in \mathbb{R}^{d_m \times (n_v \cdot d)}$, with $d_m$ as the target dimension of $z_m$.

Now, we derive the perturbation effects of $z_{cls}$ analytically to prove Proposition 2. For simplicity of derivation, we consider a single-layer attention operation. Let $\Delta z_{cls}^1$ be the perturbation applied to $z_{cls}^1$, and propagates through $z_{cls}^1 \xrightarrow{backward} T_0 \xrightarrow{forward} z_v^1$. To propagate this perturbation backward, we use the loss function of $\ell_2$ norm and get $\partial L / \partial z_{cls}^1 = z_{cls}^1$. Following the forward computation of $z_{cls}^1 = \sum_{j=0}^{n_v} A_{0,j}(T_0[j]W_V) + z_{cls}^0$, the gradients with respect to $T_0$ can be formulated as:

$$\frac{\partial z_{cls}^1}{\partial z_{cls}^0} = A_{0,0}W_V + I, \quad \frac{\partial z_{cls}^1}{\partial z_v^{0,j}} = A_{0,j}W_V, \tag{13}$$

where $I$ is the identity matrix from the residual connection. Assume $\Delta z_{cls}$ aligns with the gradient direction during the back propagation of the attack, the perturbation to $T_0$ can be:

$$
\begin{aligned}
\Delta z_{cls}^0 &= -\eta \left(\frac{\partial z_{cls}^1}{\partial z_{cls}^0}\right)^T \frac{\partial L}{\partial z_{cls}^1} = -\eta (A_{0,0}W_V + I)^T \Delta z_{cls}^1, \\
\Delta z_v^{0,j} &= -\eta \left(\frac{\partial z_{cls}^1}{\partial z_v^{0,j}}\right)^T \frac{\partial L}{\partial z_{cls}^1} = -\eta (A_{0,j}W_V)^T \Delta z_{cls}^1,
\end{aligned}
\tag{14}
$$

where $\eta > 0$ controls the magnitude of the perturbation when propagating. To compute the perturbation norms rigorously, we model the attention weights and the projection matrix. Assume the attention weights $A_{0,j}$ are approximately uniform due to similar query-key similarities in pretrained CLIP-ViT models, with $A_{0,j} = \frac{1}{n_v+1} + \epsilon_j$, $\sum_{j=0}^{n_v} \epsilon_j = 0$, $|\epsilon_j| \leq \delta_A / (n_v + 1)$, where $\delta_A$ is a small constant bounding the deviation from uniformity. Assume the projection matrix $W_V$ has bounded spectral norm, $\|W_V\|_2 \leq \sigma_V$, where $\sigma_V$ is a constant close to 1. Empirically, we compute the norm of the value projection matrix $W_V$ from the last layer of the vision encoder, yielding a value of 1.23 based on extensive experiments with the LLaVa1.5-7B model. To ensure amplitude equivalence, compute the total perturbation norm at Layer 0:

$$
\begin{aligned}
\|\Delta T_0\|_F^2 &= \|\Delta z_{cls}^0\|_2^2 + \sum_{j=1}^{n_v} \|\Delta z_v^{0,j}\|_2^2 \\
&= \eta \cdot \left(\left\|(A_{0,0}W_V + I)^T \Delta z_{cls}^1\right\|_2 + \sum_{j=1}^{n_v} \left\|(A_{0,j}W_V)^T \Delta z_{cls}^1\right\|_2\right)
\end{aligned}
\tag{15}
$$

Compute the first term in Eq. (15) as:

$$\left\|(A_{0,0}W_V + I)^T \Delta z_{cls}^1\right\|_2 = \left\|\left(\left(\frac{1}{n_v + 1} + \epsilon_0\right)W_V^T + I\right)\Delta z_{cls}^1\right\|_2$$

$$\leq \left\|\left(\frac{1}{n_v + 1} + \epsilon_0\right)W_V^T \Delta z_{cls}^1\right\|_2 + \left\|\Delta z_{cls}^1\right\|_2$$

$$\leq \left(\frac{1}{n_v + 1} + \frac{\delta_A}{n_v + 1}\right)\sigma_V \left\|\Delta z_{cls}^1\right\|_2 + \left\|\Delta z_{cls}^1\right\|_2$$

$$= \left(1 + \frac{(1 + \delta_A)\sigma_V}{n_v + 1}\right)\left\|\Delta z_{cls}^1\right\|_2$$

Similarly, compute each norm of second term in Eq. (15) as:

$$\left\|(A_{0,j}W_V)^T \Delta z_{cls}^1\right\|_2 = \left\|\left(\frac{1}{n_v + 1} + \epsilon_0\right)W_V^T \Delta z_{cls}^1\right\|_2$$

$$\leq \left(\frac{1}{n_v + 1} + \frac{\delta_A}{n_v + 1}\right)\sigma_V \left\|\Delta z_{cls}^1\right\|_2 = \frac{1 + \delta_A}{n_v + 1}\sigma_V \left\|\Delta z_{cls}^1\right\|_2$$

By combining the two terms, we can derive the equation of $\|\Delta T_0\|_F$ as:

$$\|\Delta T_0\|_F \leq \eta \cdot \left\|\Delta z_{cls}^1\right\|_2 \sqrt{\left(1 + \frac{(1 + \delta_A)\sigma_V}{n_v + 1}\right)^2 + n_v \left(\frac{(1 + \delta_A)\sigma_V}{n_v + 1}\right)^2}$$

For large $n_v$ and small constant $\delta_A$, $\sigma_V$, the dominant term is the first as:

$$\|\Delta T_0\|_F \leq \eta \cdot \left\|\Delta z_{cls}^1\right\|_2 \sqrt{1 + \frac{2(1 + \delta_A)\sigma_V + (1 + \delta_A)^2 \sigma_V^2}{n_v + 1}} = \eta \cdot \left\|\Delta z_{cls}^1\right\|_2 + \epsilon_T,$$

where $\epsilon_T$ is a small constant related to $\delta_A$ and $\sigma_V$. So we can set $\eta = 1$ to match the magnitude of $\|\Delta z_{cls}^1\|_2$. The perturbation to $T_0$ in Eq. (14) can be reformulated as:

$$\Delta z_{cls}^0 = -(A_{0,0}W_V + I)^T \Delta z_{cls}^1, \quad \Delta z_v^{0,j} = -(A_{0,j}W_V)^T \Delta z_{cls}^1 \tag{16}$$

Then, the forward propagation to $z_v^1$ with the perturbed $\tilde{T}_0 = [z_{cls}^0 + \Delta z_{cls}^0; z_v^0 + \Delta z_v^0]$ can be formulated as:

$$\Delta z_v^{1,i} = \sum_{j=0}^{n_v} A_{i,j}(\Delta T_0[j]W_V) + \Delta T_0[i] = A_{i,0}(\Delta z_{cls}^0 W_V) + \sum_{j=1}^{n_v} A_{i,j}(\Delta z_v^{0,j}W_V) + \Delta z_v^{0,i},$$

where $z_v^{1,i}$ denotes the embedding of $i$-th token in Layer 1, while $z_v^{0,i}$ denotes the embedding of $i$-th token in Layer 0. Now we plug in the expressions of Eq. (16) and get the perturbation as:

$$\Delta z_v^{1,i} = -A_{i,0}\left[(A_{0,0}W_V + I)^T \Delta z_{cls}^1 W_V\right] - \sum_{j=1}^{n_v} A_{i,j}[(A_{0,j}W_V)^T \Delta z_{cls}^1 W_V] - (A_{0,i}W_V)^T \Delta z_{cls}^1$$

$$= -\left[A_{i,0}(A_{0,0}W_V + I)^T W_V + \sum_{j=1}^{n_v} A_{i,j}(A_{0,j}W_V)^T W_V + (A_{0,i}W_V)^T\right]\Delta z_{cls}^1 \tag{17}$$

Define $M_i = A_{i,0}(A_{0,0}W_V + I)^T W_V + \sum_{j=1}^{n_v} A_{i,j}(A_{0,j}W_V)^T W_V + (A_{0,i}W_V)^T$, then we can simplify the perturbation as $\Delta z_v^{1,i} = -M_i \Delta z_{cls}^1$. For all patch tokens, we can aggregate as:

$$\Delta z_v^1 = [\Delta z_v^{1,1}; \cdots; \Delta z_v^{1,n_v}] = -\begin{bmatrix} M_1 \\ \vdots \\ M_{n_v} \end{bmatrix}\Delta z_{cls}^1$$

Now, we need to analyze the perturbation magnitude relationship between $\Delta z_v^1$ and $\Delta z_{cls}^1$ by calculating the coefficient $M_i$. Compute the first term of $\|M_i\|_2$ as:

$$\|A_{i,0}(A_{0,0}W_V + I)^T W_V\|_2 \leq |A_{i,0}| \cdot \|(A_{0,0}W_V + I)^T W_V\|_2$$

$$\leq |A_{i,0}| \cdot \left(\|(A_{0,0}W_V)^T\|_2 + \|I\|_2\right) \cdot \|W_V\|_2$$

$$\leq \frac{1 + \delta_A}{n_v + 1} \cdot \left(1 + \frac{(1 + \delta_A)\sigma_V}{n_v + 1}\right)\sigma_V$$

Compute each term in the summation of the second term of $\|M_i\|_2$ as:

$$\left\|A_{i,j}(A_{0,j}W_V)^T W_V\right\|_2 \le |A_{i,j}| \cdot |A_{0,0}| \cdot \|W_V^T\|_2 \|W_V\|_2 \le \left(\frac{1+\delta_A}{n_v+1}\right)^2 \cdot \sigma_V^2$$

Compute the third terms of $\|M_i\|_2$ as:

$$\left\|(A_{0,i}W_V)^T\right\|_2 = \left\|W_V^T A_{0,i}\right\|_2 \le \left\|W_V^T\right\|_2 \cdot |A_{0,i}| \le \sigma_V \cdot \frac{1+\delta_A}{n_v+1}$$

Combine the three terms to get the norm of $M_i$ and simplify for large $n_v$ as:

$$
\begin{aligned}
\|M_i\|_2 &\le \frac{1+\delta_A}{n_v+1} \cdot \left(1 + \frac{(1+\delta_A)\,\sigma_V}{n_v+1}\right)\sigma_V + \frac{n_v\,(1+\delta_A)^2\,\sigma_V^2}{(n_v+1)^2} + \frac{1+\delta_A}{n_v+1}\sigma_V \\
&= \frac{(1+\delta_A)\,\sigma_V}{n_v+1} \cdot \left(1 + \frac{(1+\delta_A)\,\sigma_V}{n_v+1} + 1\right) + \frac{(1+\delta_A)^2\,\sigma_V^2}{n_v+1} \quad\quad (18)\\
&= \frac{2\,(1+\delta_A)\,\sigma_V + (1+\delta_A)^2\,\sigma_V^2}{n_v+1}
\end{aligned}
$$

Substituting the Eq. (18) into Eq. (17), the norm of $\Delta z_v^{1,i}$ can be:

$$\left\|\Delta z_v^{1,i}\right\|_2 = \left\|M_i \Delta z_{cls}^1\right\|_2 \le \|M_i\|_2 \cdot \left\|\Delta z_{cls}^1\right\|_2 \le \frac{2\,(1+\delta_A)\,\sigma_V + (1+\delta_A)^2\,\sigma_V^2}{n_v+1} \cdot \left\|\Delta z_{cls}^1\right\|_2$$

Thus the Frobenius norm of $\Delta z_v^1$ is:

$$
\begin{aligned}
\left\|\Delta z_v^1\right\|_F &= \sqrt{\sum_{i=1}^{n_v}\left\|\Delta z_v^{1,i}\right\|_2^2} \le \sqrt{\sum_{i=1}^{n_v}\left(\frac{2\,(1+\delta_A)\,\sigma_V + (1+\delta_A)^2\,\sigma_V^2}{n_v+1} \cdot \left\|\Delta z_{cls}^1\right\|_2\right)^2} \\
&= \sqrt{n_v} \cdot \frac{2\,(1+\delta_A)\,\sigma_V + (1+\delta_A)^2\,\sigma_V^2}{n_v+1} \cdot \left\|\Delta z_{cls}^1\right\|_2 \quad\quad (19)\\
&= \frac{2\,(1+\delta_A)\,\sigma_V + (1+\delta_A)^2\,\sigma_V^2}{\sqrt{n_v}} \cdot \left\|\Delta z_{cls}^1\right\|_2
\end{aligned}
$$

As $\delta_A$ is a small constant bounding the deviation from uniformity, $\sigma_V$ is a bounded spectral norm of $\|W_V\|_2$ and close to 1, we can simplify the coefficients in Eq. (19) as $\frac{3+\epsilon_V}{\sqrt{n_v}}$, where $\epsilon_V \ll 1$ and is related to the value of $\delta_A$ and $\sigma_V$. Then the ratio of amplitude through $z_{cls}^1 \xrightarrow{backward} T_0 \xrightarrow{forward} z_v^1$ can be bounded as:

$$\frac{\left\|\Delta z_v^1\right\|_F}{\left\|\Delta z_{cls}^1\right\|_2} \le \frac{3+\epsilon_V}{\sqrt{n_v}}, \quad \epsilon_V \ll 1 \quad\quad (20)$$

This matches the first claim of Proposition 2. Now, we prove the second view. For the same degree of perturbation on $z_{cls}$ or $z_v$, we can assume $\|\Delta z_{cls}^1\|_2 = \Delta$ and $\|\Delta z_v^1\|_F = \Delta$. Then we will compare the effect of adding perturbation $\Delta z_{cls}$ or $\Delta z_v$.

If the feature of attack objective is $z_{cls}$, following the derivation in Section B, we have:

$$\sigma_{\min}(W_a)\,\|\Delta z_v(z_{cls})\|_F \le \|\Delta z_m(z_{cls})\|_F \le \sigma_{\max}(W_a)\,\|\Delta z_v(z_{cls})\|_F \quad\quad (21)$$

If the feature of attack objective is $z_v$, the bound of $z_m$ in this context can also be:

$$\sigma_{\min}(W_a)\,\|\Delta z_v(z_v)\|_F \le \|\Delta z_m(z_v)\|_F \le \sigma_{\max}(W_a)\,\|\Delta z_v(z_v)\|_F \quad\quad (22)$$

Combining Eq. (21) and (22), as well as conditions in Eq. (20), we can obtain that under the same degree of perturbation, the ratio of the lower bound of $\Delta z_m$ is:

$$\frac{\|\Delta z_m(z_{cls})\|_F}{\|\Delta z_m(z_v)\|_F} = \frac{\sigma_{\min}(W_a)\,\|\Delta z_v(z_{cls})\|_F}{\sigma_{\min}(W_a)\,\|\Delta z_v(z_v)\|_F} \le \frac{((3+\epsilon_V)/\sqrt{n_v})\cdot\|z_{cls}^1\|_2}{\|\Delta z_v^1\|_F} = \frac{3+\epsilon_V}{\sqrt{n_v}}$$

This matches the second claim of Proposition 2. When extended to $L$-layer CLIP, the above proof occurs between $L$ and $L-1$ layers, where the propagation is in the form of $[z_{cls}^L + \Delta z_{cls}^L; z_v^L] \to [z_{cls}^{L-1} + \Delta z_{cls}^{L-1}; z_v^{L-1} + \Delta z_v^{L-1}] \to [z_{cls}^L; z_v^L + \Delta z_v^L]$. Before $L-1$ layer, the propagation follows the $[z_{cls}^i + \Delta z_{cls}^i; z_v^i + \Delta z_v^i] \to [z_{cls}^{i-1} + \Delta z_{cls}^{i-1}; z_v^{i-1} + \Delta z_v^{i-1}]$ propagation method. Since we have assumed the amplitude equivalence before and after the propagation, we simplify the propagation of the first $L-1$ layer as having the same amplitude. Therefore, our conclusion still holds.

## D    MORE RESULTS ON MULTI-TASK TRANSFER ATTACK

In Fig. 2(b) of the main text, we initially demonstrated the transfer attack between COCO Caption and VQAv2 datasets, representing two distinct downstream tasks, and confirmed that VEAttack effectively achieves downstream-agnostic attacks across these tasks. Table 6 provides a more detailed and comprehensive analysis of multi-task transfer attack performance. We use "**Anytask**" to denote that VEAttack targets any task, such as COCO Caption or VQA, as it solely attacks the vision encoder, enabling downstream-agnostic performance that remains consistent across different tasks. For APGD, we underline the performance of transfer attacks between tasks, revealing that these reductions (*e.g.*, 49.4% for VQAv2 to COCO Caption, and 29.2% for VQAv2 to OK_VQA) are consistently less severe than those achieved by VEAttack (*e.g.*, 93.9% for COCO Caption, and 62.8% for OK_VQA). This highlights the limitations of traditional white-box attacks like APGD in multi-task LVLM scenarios, while further validating the effectiveness and robustness of VEAttack in achieving significant performance degradation across diverse tasks.

Table 6: Transfer Attack Performance of APGD and VEAttack Between Captioning and VQA Tasks. The underlined values in APGD represent transfer attack results, and diagonal values indicate direct white-box attack performance.

| Attack | Target / Source | COCO Caption | VQAv2 | OK_VQA |
|---|---|---|---|---|
| | Clean | 115.5 | 74.5 | 58.9 |
| APGD | COCO Caption | 13.1 | 45.9 | 25.7 |
| | VQAv2 | 58.5 | 25.3 | 41.7 |
| | OK_VQA | 43.8 | 53.3 | 14.6 |
| VEAttack | **Anytask** | **7.1** | **38.4** | **21.9** |

## E    EFFECTS AND COST OF VEATTACK WITH LLM

To further demonstrate whether the effect of attacking the vision encoder is sufficient, we add a joint attack on LLM to the current VEAttack, where we supervised all the hidden states of LLM outputs. The results in the Table 7 demonstrate a further enhancement in attack effectiveness on the Flickr30k dataset when attacking jointly the vision encoder and LLM, with a performance gain of 0.6. However, this improvement comes at a significant computational cost, increasing runtime by 278%. Through this observation, LLM shows a limited effect in the attack but highly increases the computational cost, further validating the necessity of focusing on vision encoder attacks as proposed in our framework.

Table 7: Comparison of VEAttack and VEAttack+LLM backbone attack on LLaVa1.5-7B.

| Method | Dataset | LLaVa1.5-7B $\epsilon = 2/255$ | $\epsilon = 4/255$ | Time |
|---|---|---|---|---|
| VEAttack | COCO | **10.8** | **7.1** | **51min38s** |
| | Flickr30k | 10.7 | 6.3 | **49min59s** |
| VEAttack+LLM | COCO | 12.3 | 7.3 | 3h11min7s |
| | Flickr30k | **10.1** | **5.4** | 3h8min58s |

## F    MORE EVALUATION MODELS AND METRICS ON IMAGE CAPTIONING

Table 8 presents a comprehensive evaluation of the VEAttack method on image captioning across the COCO and Flickr30k datasets using the LLaVa1.5-7B model. Based on Eq. 5, we incorporate attacks from adversarial training algorithms (Mao et al., 2023) as proposed in the robust CLIP framework, as a baseline method to enable a more comprehensive comparison. The formulation can be expressed as $\mathcal{L}_{cls}$ : $\tilde{v} = \arg\max_{\|\delta\| \leq \epsilon} -\mathbb{E}_j \left[ \cos\left(\tilde{z}_{cls}, z_{text}^j\right) \right]$, where text features $z_{text}^j$ is extracted from

user instructions. Compared to the clean baselines, VEAttack consistently demonstrates the most significant performance degradation on COCO captioning across all evaluation metrics. Specifically, BLEU@4, which focuses on measuring caption precision, shows a substantial drop of $86.7\%$ on the COCO dataset, indicating the highly effective nature of our VEAttack. ROUGE_L, which evaluates recall and sequence overlap, declines moderately $45.8\%$, suggesting that while the attack disrupts overall performance, some structural coherence in the captions remains. SPICE, focusing on semantic propositional content, drops $87.1\%$, indicating a broad degradation in meaningful content representation, reinforcing the effectiveness of VEAttack.

Table 8: More evaluation metrics for image captioning across datasets and attack methods.

| Datasets | Attack | BLUE@4 | METEOR | ROUGE_L | CIDEr | SPICE |
|---|---|---|---|---|---|---|
| COCO | Clean | 33.0 | 28.3 | 56.6 | 115.5 | 22.4 |
| | $L_{cls}$ Mao et al. (2023) | 14.8 | 17.5 | 40.6 | 41.8 | 9.9 |
| | $L_2$ Schlarmann et al. (2024) | 10.6 | 15.0 | 36.7 | 25.2 | 7.3 |
| | QAVA Zhang et al. (2025) | 22.3 | 24.7 | 49.2 | 79.9 | 17.8 |
| | VEAttack | **4.4** | **10.9** | **30.7** | **7.1** | **2.9** |
| Flickr30k | Clean | 29.6 | 25.2 | 52.6 | 77.5 | 18.2 |
| | $L_{cls}$ Mao et al. (2023) | 15.5 | 17.0 | 39.5 | 29.9 | 9.8 |
| | $L_2$ Schlarmann et al. (2024) | 12.9 | 14.8 | 36.3 | 20.6 | 7.8 |
| | VEAttack | **6.6** | **10.6** | **30.6** | **6.3** | **4.2** |

To demonstrate the broad applicability of VEAttack beyond CLIP-based architectures, we extend our evaluation to LVLMs utilizing diverse vision encoders. Specifically, we test MiniGPT-4 (Zhu et al., 2023), which employs the BLIP-2 (Li et al., 2023a) as a vision encoder, and mPLUG-Owl2 (Ye et al., 2024), which uses a distinct visual backbone, MplugOwl. We compare VEAttack against two gray-box methods, AttackVLM-ii (Zhao et al., 2023) and VT-Attack (Wang et al., 2024c). Following the setting of VT-Attack, we set the perturbation budget $\epsilon = 8/255$, step size $\alpha = 1/255$, and attack iterations $t = 100$. As shown in Table 9, VEAttack consistently outperforms the baselines, achieving the most significant performance degradation on both models. On MiniGPT-4, VEAttack reduces the CIDEr score to 13.7, compared to 24.8 for VT-Attack. Similarly, on mPLUG-Owl2, our method achieves a CIDEr score of 65.1, significantly lower than the other gray-box methods. These results confirm that VEAttack effectively disrupts the visual semantic representation regardless of the underlying vision encoder architecture, validating its architecture-agnostic nature.

Table 9: Attack performance comparison on COCO Captioning against more LVLMs.

| Models | Attack | BLUE@4 | METEOR | ROUGE_L | CIDEr | SPICE |
|---|---|---|---|---|---|---|
| MiniGPT-4 | Clean | 25.0 | 27.9 | 53.1 | 94.0 | 21.3 |
| | AttackVLM-ii Zhao et al. (2023) | 5.8 | 13.5 | **30.0** | **5.4** | 5.4 |
| | VT-Attack Wang et al. (2024c) | 16.3 | 21.9 | 44.5 | 54.9 | 14.2 |
| | VEAttack | **5.1** | **12.3** | 30.5 | 10.0 | **4.5** |
| mPLUG-Owl2 | Clean | 38.6 | 30.2 | 59.5 | 139.2 | 24.3 |
| | AttackVLM-ii Zhao et al. (2023) | 20.6 | 21.0 | 46.8 | 71.6 | 13.8 |
| | VT-Attack Wang et al. (2024c) | 17.7 | 19.2 | 43.9 | 59.8 | 12.5 |
| | VEAttack | **12.5** | **15.8** | **38.8** | **40.4** | **8.3** |

# G  ATTACK PERFORMANCE ON IMAGE-TEXT RETRIEVAL TASK

## G.1  ATTACK PERFORMANCE UNDER WHITE-BOX OF VISION ENCODER

We evaluate VEAttack on image–text retrieval using CLIP-ViT and compare against SGA (Lu et al., 2023) and DRA (Gao et al., 2024) on Flickr30k in Table 10. Following prior work, we use Attack Success Rate (ASR) as the primary measure of transferability. In this setting, we report R@1 and R@10 for both Text Retrieval (TR) and Image Retrieval (IR), where each R@k is interpreted as ASR@k, the fraction of adversarial queries whose ground-truth counterpart does not appear among the top-k retrieved results. As shown in the table, VEAttack attains competitive ASR, while being substantially more efficient, reducing time to 10min38s and memory to 3.13 GB. These results

indicate that our vision-encoder–only optimization delivers strong retrieval-level attack efficacy with markedly lower computational cost.

Table 10: Comparison of SGA, DRA and VEAttack on the Image-Text Retrieval task.

| Method | TR R@1 | TR R@10 | IR R@1 | IR R@10 | Time | Memory |
|--------|--------|---------|--------|---------|------|--------|
| SGA Lu et al. (2023) | **100** | **99.9** | **100** | **99.98** | 14min17s | 9.14GB |
| DRA Gao et al. (2024) | **100** | **99.9** | **100** | 99.96 | 26min31s | 10.31GB |
| VEAttack | 99.88 | 99.19 | 99.81 | 98.54 | **10min38s** | **3.13GB** |

### G.2 TRANSFER ATTACK PERFORMANCE OF VISION ENCODERS

We additionally evaluate the transfer attack performance following the transfer-attack methods, SGA (Lu et al., 2023) and DRA (Gao et al., 2024). We conduct transfer attacks using CLIP-ViT as the source model to attack ALBEF and CLIP-CNN. As shown in Table 11, while SGA and DRA are specifically designed with complex augmentation strategies to maximize transferability, VEAttack achieves competitive performance with fewer computational memory costs. Crucially, leveraging our Möbius Band in Observation 3, we replace the standard CLIP-ViT source with the robust FARE encoder. This simple substitution dramatically boosts VEAttack's transferability, requiring no complex algorithm changes but achieving an R@1 of $35.35\%$ on ALBEF (Text Retrieval) and $48.78\%$ (Image Retrieval). This performance rivals the sophisticated DRA method while consuming only $10.52$GB of memory. This demonstrates that utilizing robust vision encoders is a highly efficient and effective strategy for transfer attacks, offering a superior trade-off between performance and resource consumption.

Table 11: Comparison of transfer attack with SGA and DRA on the Image-Text Retrieval task.

| Method | Source | ALBEF (TR R@1) | ALBEF (IR R@1) | CLIP-CNN (TR R@1) | CLIP-CNN (IR R@1) | Memory |
|--------|--------|----------------|----------------|-------------------|-------------------|--------|
| SGA | CLIP-ViT | 22.42 | 34.59 | 53.26 | 61.1 | 16.63GB |
| DRA | CLIP-ViT | 27.84 | 42.84 | **64.88** | **69.50** | 16.71GB |
| VEAttack | CLIP-ViT | 16.58 | 32.23 | 44.06 | 53.55 | **6.75GB** |
| | FARE | **35.35** | **48.78** | 55.17 | 63.05 | 10.52GB |

## H ATTACK PERFORMANCE ON VISUAL GROUNDING TASK

To verify the generalizability of VEAttack, we extended our evaluation to visual grounding task and select ReCLIP (Subramanian et al., 2022), a popular zeroshot baseline for visual grounding that relies on the CLIP vision encoder to localize objects. We evaluated the attack performance on the RefCOCO and RefCOCO+ datasets (Yu et al., 2016).

Table 12: Attack performance after difference methods on Visual Grounding task.

| Method | RefCOCO | | | RefCOCO+ | | |
|--------|---------|-------|-----|----------|-------|-----|
| | TestA | TestB | Val | TestA | TestB | Val |
| Clean | 49.0 | 53.9 | 48.2 | 50.8 | 49.1 | 47.3 |
| AttackVLM-ii | 27.6 | 33.5 | 28.9 | 28.5 | **27.9** | 29.1 |
| VT-Attack | 35.4 | 40.7 | 34.5 | 34.7 | 37.9 | 35.7 |
| VEAttack | **23.1** | **32.1** | **27.1** | **24.6** | 29.6 | **27.4** |

For the attack settings, we maintain consistency across all comparison methods by setting the perturbation budget to $\epsilon = 2/255$ and the number of iterations to $t = 100$. As shown in Table 12, VEAttack exhibits effectiveness in disrupting the spatial reasoning capabilities. On the RefCOCO validation set, VEAttack degrades the performance to 27.1, significantly outperforming the baselines. This demonstrates that our vision-encoder-only attack effectively propagates to spatial localization tasks, further validating its downstream-agnostic nature.

## I TRANSFER ATTACK FROM LARGER SPECIFIC VISION ENCODERS

We conducted transfer attacks using the vision encoders of mPLUG-Owl2 and Qwen-VL as source models on 1000 random images of COCO dataset with the CIDEr metric of the captioning task in Table 13 with $\epsilon = 16/255$, $\alpha = 16/(255 \times 5)$. Surprisingly, we found that for both VEAttack and VT-Attack, when using the vision encoders of mPLUG-Owl2 and Qwen-VL as source models for transfer attacks, it is challenging to achieve successful attacks on other models. One potential reason

could be that models tend to develop more specialized feature representations tailored to the specific LVLM architecture and training distribution as the scale of training data and network complexity increases. It might reduce the transferability of adversarial perturbations across diverse model architectures. In addition, the generated adversarial perturbations from these large vision encoders are concentrated in high-dimensional feature spaces. When transferred to smaller models such as CLIP, these perturbations may fail to effectively decouple or disrupt downstream task performance, as the limited capacity of smaller models may hinder their ability to process or adapt to such complex disturbances. This finding provides an empirical explanation for the community's current practice of using an ensemble of foundational vision encoders like CLIP for attacks (Zhao et al., 2023; Zhang et al., 2024; Li et al., 2025; Jia et al., 2025).

Table 13: Transfer attack on COCO captioning from mPLUG-Owl2 and Qwen-VL to other models.

| Method | Source | Target | CIDEr | Source | Target | CIDEr |
|---|---|---|---|---|---|---|
| VEAttack | mPLUG-Owl2 | mPLUG-Owl2 | **9.1** | Qwen-VL | Qwen-VL | **19.3** |
| | | Qwen-VL | 131.8 | | mPLUG-Owl2 | 128.3 |
| | | LLaVa1.5-7B | 114.5 | | LLaVa1.5-7B | 119.4 |
| | | OpenFlamingo-9B | 84.5 | | OpenFlamingo-9B | 86.0 |
| VT-Attack | mPLUG-Owl2 | mPLUG-Owl2 | 30.1 | Qwen-VL | Qwen-VL | 55.3 |
| | | Qwen-VL | 132.1 | | mPLUG-Owl2 | 130.8 |
| | | LLaVa1.5-7B | 119.6 | | LLaVa1.5-7B | 121.5 |
| | | OpenFlamingo-9B | 91.6 | | OpenFlamingo-9B | 96.3 |

**Analysis on feature representation.** To delve deeper into the underlying causes of the limited transferability observed in Table 13, we visualize the feature distributions of different vision encoders. Specifically, we randomly sample 100 patch tokens from the COCO validation set encoded by the vision encoders of LLaVa1.5-7B (CLIP-ViT-L), mPLUG-Owl2 (MplugOwl), and Qwen-VL (CLIP-ViT-bigG). As illustrated in Fig. 9, the t-SNE visualization reveals significant distributional discrepancies among these models. The feature spaces of these encoders form distinct, isolated clusters with large margins, indicating that the specific vision encoders used in large-scale LVLMs undergo substantial specific adaptation or fine-tuning, leading

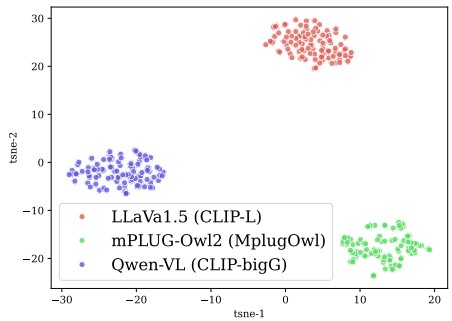

Figure 9: t-SNE visualization of visual feature distributions across different LVLMs.

to a severe feature shift compared to standard foundation models like CLIP. Consequently, adversarial perturbations optimized on the specific manifold of one large encoder (*e.g.*, Qwen-VL) are likely orthogonal or irrelevant to the feature space of other models (*e.g.*, LLaVa).

## J  EVALUATION ON IMAGE CLASSIFICATION DATASETS

Since the CLIP can also function as a classifier, we assess the impact of VEAttack across diverse classification datasets, including Tiny-ImageNet (Deng et al., 2009), CIFAR-10 (Krizhevsky et al., 2009), CIFAR-100 (Krizhevsky et al., 2009), FOOD101 (Bossard et al., 2014), Flowers102 (Nilsback & Zisserman, 2008), DTD (Cimpoi et al., 2014), EuroSAT (Helber et al., 2019), FGVC-Aircraft (Maji et al., 2013), Caltech-256 (Griffin et al., 2007), StanfordCars (Krause et al., 2013), ImageNet (Deng et al., 2009) and SUN397 (Xiao et al., 2010). Table 14 presents the evaluation of VEAttack on image classification datasets using the CLIP-B/32 model. The results show that $\mathcal{L}_{cls}$, which optimizes the cross-entropy loss between classification and text features, achieves the most significant performance degradation of $97.8\%$, as its objective aligns closely with the classification task. Compared with the $\mathcal{L}_2$ attack, which uses the $\ell_2$ loss of class token embedding as the attack objective, VEAttack achieves a more substantial reduction in mean accuracy of $93.0\%$, demonstrating its superior efficacy in disrupting classification performance. This improvement also highlights the advantage of perturbing patch tokens over class tokens, underscoring its effectiveness over traditional methods.

Table 14: Performance evaluation of image classification datasets under different attacks.

| Attack | | Tiny-ImageNet | CIFAR-10 | CIFAR-100 | FOOD101 | Flowers102 | DTD | EuroSAT | FGVC-Aircraft | Caltech-256 | StanfordCars | ImageNet | SUN397 | Mean |
|---|---|---|---|---|---|---|---|---|---|---|---|---|---|---|
| Clean | | 57.9 | 88.1 | 60.5 | 83.8 | 65.7 | 40.1 | 38.2 | 20.2 | 82.0 | 52.1 | 59.1 | 57.7 | 58.8 |
| $\mathcal{L}_2$ Schlarmann et al. (2024) | $\epsilon = 4/255$ | 4.9 | 17.3 | 5.1 | 12.2 | 16.8 | 11.5 | 15.4 | 4.1 | 25.5 | 9.9 | 12.1 | 12.6 | 12.3 |
| | $\epsilon = 8/255$ | 1.6 | 12.7 | 3.0 | 2.8 | 5.3 | 6.4 | 13.1 | 1.6 | 10.8 | 1.9 | 3.8 | 4.1 | 5.6 |
| VEAttack | $\epsilon = 4/255$ | 2.6 | 15.3 | 3.2 | 5.9 | 8.8 | 10.6 | 13.2 | 2.0 | 11.9 | 5.8 | 4.7 | 3.4 | 7.3 |
| | $\epsilon = 8/255$ | 1.0 | 13.8 | 2.2 | 1.5 | 2.9 | 7.0 | 13.3 | 1.2 | 3.9 | 0.8 | 1.1 | 0.7 | 4.1 |
| $\mathcal{L}_{cls}$ Mao et al. (2023) | $\epsilon = 4/255$ | 0.5 | 2.1 | 0.1 | 5.4 | 0.4 | 0.0 | 0.0 | 0.0 | 6.8 | 0.1 | 0.4 | 0.3 | 1.3 |

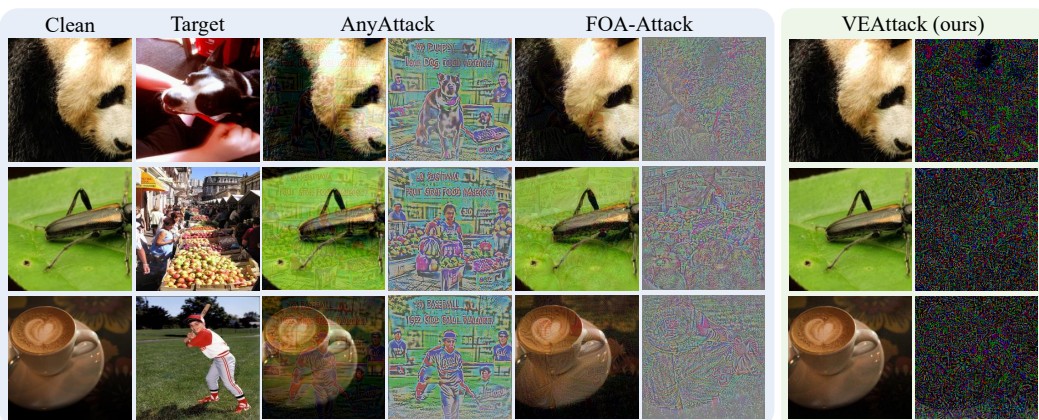

Figure 10: Comparison with perturbation of targeted black-box attacks AnyAttack (Zhang et al., 2024) and FOA-Attack (Jia et al., 2025).

# K  QUALITATIVE RESULTS OF VEATTACK

## K.1  COMPARISON WITH PERTURBATION OF BLACK BOX ATTACKS

Fig. 10 shows the adversarial images generated by our VEAttack, and two targeted black-box attacks, AnyAttack (Zhang et al., 2024) and FOA-Attack (Jia et al., 2025). Visual inspection shows that VEAttack yields substantially higher imperceptibility. For example, in the second row, the adversarial results of AnyAttack and FOA-Attack imprint recognizable fruit contours from the target image onto the green leaf, making the manipulation conspicuous, whereas the VEAttack image remains nearly indistinguishable from the clean input. To make this contrast explicit, we also visualize the perturbations. VEAttack achieves effective LVLM degradation with low-magnitude, semantically unstructured noise, indicating that VEAttack could attain stronger perceptual stealth.

## K.2  MORE COMPARISONS OF ADVERSARIAL IMAGES

To demonstrate the superior imperceptibility of VEAttack, we conduct a detailed qualitative and quantitative comparison with the black-box M-Attack (Li et al., 2025), as shown in Figure 11. Our goal regarding imperceptibility is to achieve effective attacks using significantly smaller perturbations compared to black-box methods, which often require heavy noise to ensure transferability. Visually, M-Attack introduces conspicuous texture distortions. Notably, it tends to generate perturbations containing semantic information; for instance, in the fourth column of the first row, word-shaped artifacts are clearly visible in the sky. In contrast, VEAttack generates perturbations that are texture-agnostic and devoid of semantic artifacts, making them much harder for human observers to detect. Quantitatively, we evaluated 500 randomly sampled adversarial images. VEAttack achieves an average $L_2$ distance of 4.3883 between the adversarial and clean images, whereas M-Attack

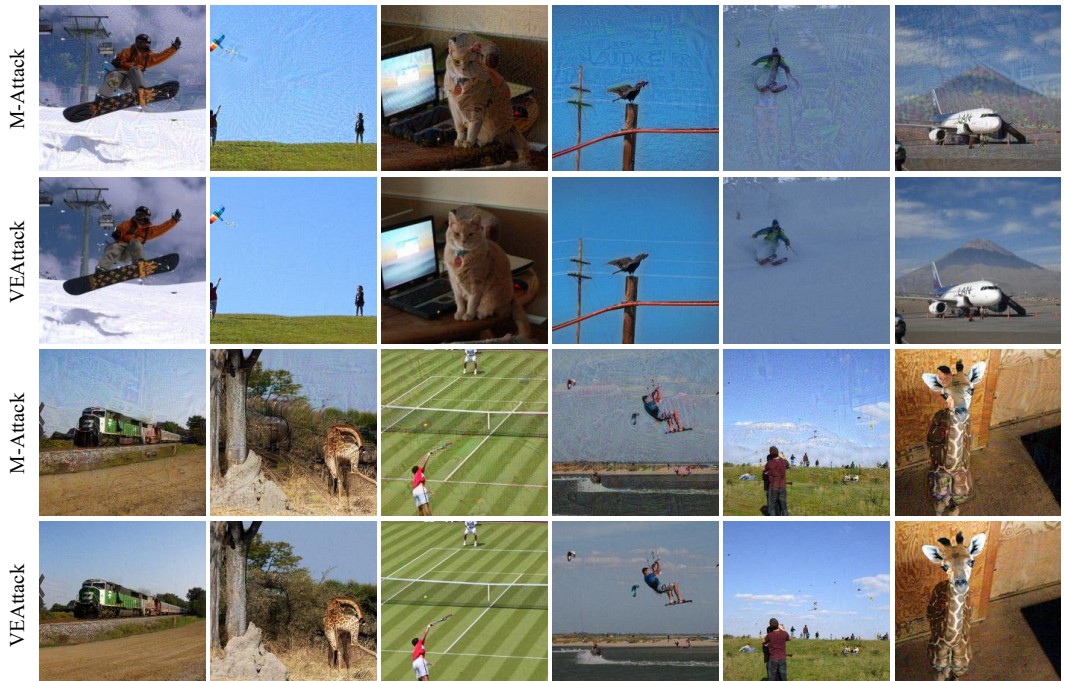

Figure 11: Visual comparison of adversarial examples generated by the black-box M-Attack (Li et al., 2025) and our gray-box VEAttack.

yields a much higher distance of 15.3403. Furthermore, we calculate the cosine similarity (CLIP Score) between the CLIP-L/14 features of the clean and adversarial images. VEAttack achieves a score of 0.4443, compared to 0.5493 for M-Attack. This indicates that VEAttack successfully pushes the visual features further away from the original semantics, while maintaining a much lower visual perturbation budget than the black-box attacks.

### K.3 QUALITATIVE RESULTS ON IMAGE CAPTIONING

We visualize the outputs of clean samples, those attacked by the traditional white-box APGD method (Croce & Hein, 2020), and our proposed VEAttack, as shown in Fig. 12. Both attack methods cause errors in the image captioning outputs of LVLMs. However, an interesting observation emerges: the traditional white-box attack, which targets output tokens as its objective, tends to generate errors that still revolve around the original sentence structure or subject, such as "in the water" in the first image, "running" in the second, and "spraying water" in the third. In contrast, since VEAttack operates in a downstream-agnostic context, it directly disrupts the semantic information in visual features of the vision encoder, leading to outputs that completely deviate from the clean captions, for example, "table, bear" in the first image, "food" in the second, and "game controller" in the third. While recent work (Schlarmann & Hein, 2023) mitigates such issues through targeted attacks to improve effectiveness, our non-targeted VEAttack approach successfully achieves a complete semantic deviation under non-targeted conditions.

### K.4 QUALITATIVE RESULTS ON VISUAL QUESTION ANSWERING

We visualize the outputs of LVLMs on the visual question answering task under clean, APGD, and VEAttack conditions, as shown in Fig. 13. Notably, VEAttack, despite operating in a downstream-agnostic context without instruction-specific guidance, effectively disrupts visual features, leading to erroneous responses from the model in most cases. This highlights its robust attack capability. We also present a failure case in the last column, where VEAttack does not specifically alter the semantic information related to the brand monitor. Overall, VEAttack demonstrates exceptional attack effectiveness in a downstream-agnostic manner.

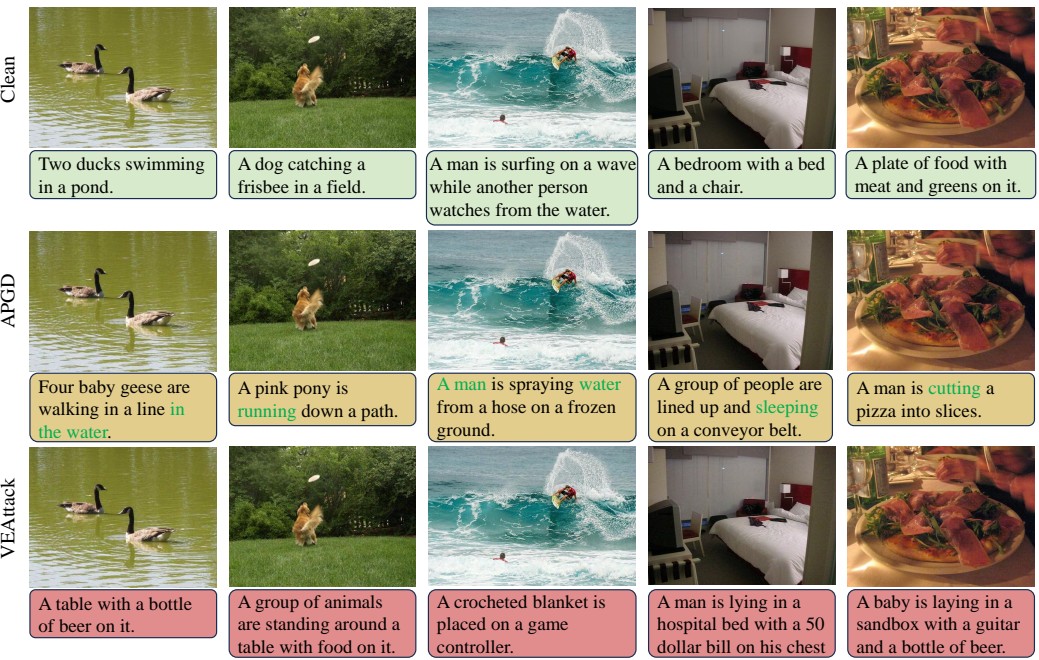

Figure 12: Quantitative results on image captioning of clean samples, traditional white-box APGD attack, and VEAttack on LLaVa1.5-7B. The green background indicates correct outputs for clean samples, the yellow background represents APGD attack outputs with green text highlighting semantically consistent content, and the red background denotes VEAttack outputs with significantly altered semantics.

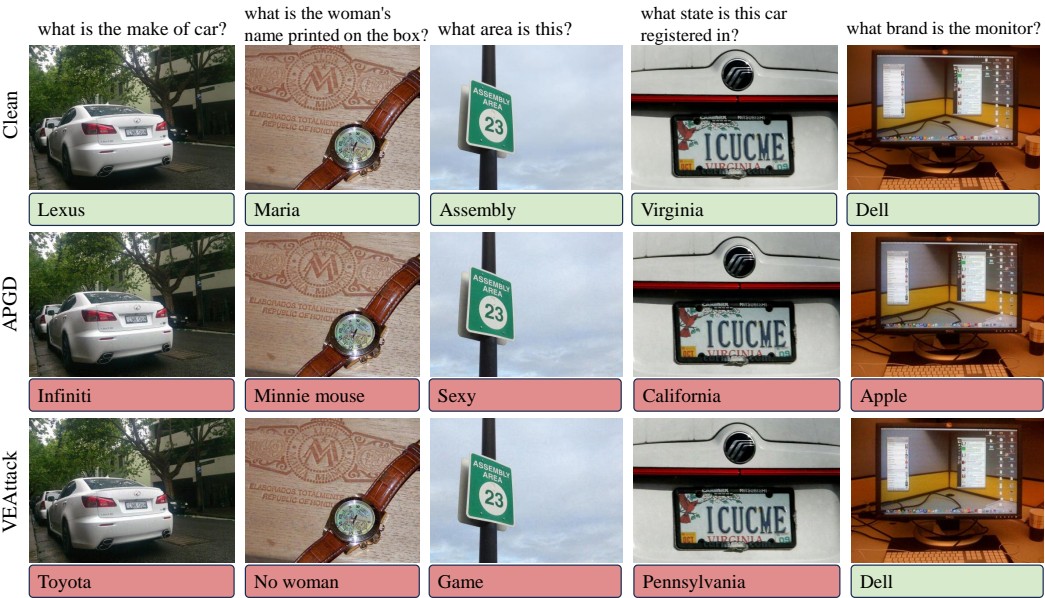

Figure 13: Quantitative results on visual question answering of clean samples, traditional white-box APGD attack, and VEAttack on LLaVa1.5-7B. The top row of each image presents the instructions posed to the LVLMs, with a green background indicating a correct answer and a red background indicating an incorrect answer.

## L  THE USE OF LARGE LANGUAGE MODELS

In accordance with the policies on LLM usage, we used a large language model as a language-refinement assistant during manuscript preparation. The use of the LLM was confined strictly to editorial support and did not contribute to the conceptual or experimental aspects of this research. Specifically, the LLM was employed for:

- Rephrase sentences and paragraphs to improve clarity, concision, and academic tone when describing our gray-box, vision-encoder-only attack, its objectives, and empirical observations (*e.g.*, the shift from class-token to patch-token perturbations and the cosine-similarity objective)

- Standardize terminology and notation across sections (*e.g.*, downstream-agnostic, untargeted, limited perturbation budget, and the symbols in Eqs. (4)–(7))

- Smooth local transitions (*e.g.*, from the redefined objective in Section 4.1 to the loss design in Section 4.2) and polish captions for clarity.

Our research ideas, technical methods, proofs, experiments, figures, and conclusions are entirely the work of the human authors. All LLM-suggested text was reviewed line-by-line and edited by the authors. Any technical statements, equations, or quantitative claims appearing in the paper were verified against our experiments and, where appropriate, cross-checked with the paper's own tables and figures.

