# OpenReview forum: "VEAttack: Downstream-agnostic Vision Encoder Attack against Large Vision Language Models"
_ICLR.cc/2026/Conference — ICLR 2026 Poster_

### Official Review · Reviewer_eK4n · 2025-10-24

**Soundness:** 3
**Presentation:** 4
**Contribution:** 4
**Rating:** 8
**Confidence:** 4

**Summary:**

This paper proposes VEAttack, a simple yet effective gray-box attack on LVLMs. VEAttack generates adversarial examples by perturbing solely the image token features of the vision encoder. The results conduct evaluations of multiple LVLMs across Visual Question Answering (VQA) and image captioning.

**Strengths:**

1. The paper is clearly structured and articulately written, ensuring ease of understanding.

2. The study demonstrates detailed analysis and keen observation.

3. Experiments were conducted across a varied set of datasets and models.

**Weaknesses:**

**Clarity:**

1. The Introduction mainly introduces the figures in other chapters, but there is no place in the paper to introduce Figure 1.

**Experiment:**

2. The authors clearly demonstrate their motivation by comparing with white-box and black-box attacks in Figure 2, but I have some confusion about their performance: Since the white-box attack performs a full gradient backpropagation, why is the black-box attack so much more time-consuming than others? Furthermore, the black-box attack performs poorly in the untargeted attack setting. Could this be related to the black-box attacks that target a specific sample? Can the authors verify this difference with other black-box attacks, such as M-Attack [1] and higher perturbation in black-box?

3. How is the “performance after attack” in Figure 2 (b) evaluated? Why is its performance trend opposite to that in Table 1? If it’s a performance decrease, please describe it clearly and align it with Table 1.

4. The authors seem to only have some subjective comparisons in Figure 8 to compare the imperceptibility with the black-box. Using data evaluation such as L2 or CLIP score would be more convincing.

[1] Li Z, Zhao X, Wu D D, et al. A frustratingly simple yet highly effective attack baseline: Over 90% success rate against the strong black-box models of gpt-4.5/4o/o1

**Questions:**

All concerns and questions are listed in the Weakness section.

---

> ### Author Response · Authors · 2025-11-21
> **Response to Reviewer eK4n**
>
> **Dear Reviewer eK4n,**
>
> We greatly appreciate your recognition of this work. Our responses to all your concerns are as follows:
>
> **Response to Weakness 1: Introduction of Figure 1.**
>
> We thank the reviewer for catching this oversight. In the revised manuscript, we have added a reference to Figure 1 in the second paragraph of the Introduction. Specifically, we cite it when introducing the gray-box setting to visually contrast the access levels and advantages of VEAttack compared to white-box and black-box attacks.
>
> **Response to Weakness 2: Efficiency and Effectiveness of Black-box Attacks.**
>
> We deeply appreciate your professional analysis and insightful hypothesis regarding the behavior of black-box attacks. Following your suggestion, we conducted verification experiments with the black-box method M-Attack [1], using a higher perturbation budget in Table 1 below. You are correct that standard gradient backpropagation (White-box) is expensive, but black-box attacks are often even costlier. This is because they do not simply transfer a gradient, but employ complex optimization frameworks, e.g., model ensemble in [1] and optimal transport in [2], to ensure the generated examples can transfer across models. In addition, we agree with your hypothesis that black-box attacks typically rely on targeted objectives to guide the perturbation toward a specific error mode that transfers well. VEAttack causes a fundamental collapse in the visual representation by directly minimizing the cosine similarity of visual features. These results empirically validate your insight and highlight the unique value of our gray-box vision-encoder-only paradigm.
>
> **Table 1**: Comparison of computational efficiency and attack effectiveness between the black-box M-Attack
> | Method | $\epsilon$ | COCO | TextVQA | Time |
> |--|--|--|--|--|
> | M-Attack | 16/255 | 15.1 | 20.5 | 539min |
> | VEAttack | 4/255 | **7.1** | **10.1** | **51min** |
>
> **Response to Weakness 3: Performance Alignment.**
>
> We thank the reviewer for the valuable suggestion. We have updated the y-axis label to "Score Degradation Rate" to explicitly indicate the metric used. Following your suggestion, we have updated the "Average" column in Table 1 of the revised paper to also report the performance degradation rate, which ensures that the data trends in the figure and the table are now fully consistent.
>
> **Response to Weakness 4: More Evaluation of Adversarial Samples.**
>
> Thank you for the valuable and insightful comments. In the revised manuscript, we have added a new section titled "More Comparisons of Adversarial Images" (Section L.2), which includes both expanded qualitative visualizations and quantitative metrics to substantiate our claim.
>
> **(1) Quantitative Evidence.** To move beyond subjective comparison, we conducted a quantitative evaluation on 500 randomly sampled adversarial images generated by VEAttack and black-box M-Attack [1]. VEAttack achieves an average $L_2$ distance of 4.3883 between adversarial and clean images, which is lower than 15.3403 of M-Attack. This confirms that our method introduces significantly less pixel-level noise. Despite the lower perturbation budget, VEAttack achieves a CLIP similarity score of 0.4443, which is lower than the 0.5493 of  M-Attack. This indicates that VEAttack disrupts the semantic visual features more effectively while maintaining higher visual fidelity.
>
> **(2) Qualitative Comparison.** In Figure 11 of our revised paper, we visualize more comparisons of adversarial samples, indicating that VEAttack produces perturbations that are texture-agnostic and are less perceptible to human observers.
>
> We hope our responses address your concerns and look forward to your reply and discussion.
>
>
> **References**
>
> [1] Li Z, Zhao X, Wu D D, et al. A frustratingly simple yet highly effective attack baseline: Over 90% success rate against the strong black-box models of gpt-4.5/4o/o1[J]. arXiv preprint arXiv:2503.10635, 2025.
>
> [2] Jia X, Gao S, Qin S, et al. Adversarial Attacks against Closed-Source MLLMs via Feature Optimal Alignment[J]. arXiv preprint arXiv:2505.21494, 2025.

---

> > ### Comment · Reviewer_eK4n · 2025-11-25
> >
> > The new comparison with black-box attacks clarifies the efficiency and imperceptibility. The approach is well-motivated and practical. I keep my score and lean towards accept.

---

> ### Comment · Area_Chair_gUSm · 2025-11-25
>
> Dear Reviewer eK4n,
>
> The authors have responded to your reviews. Please review and provide your feedback and responses.
>
> Best,
>
> Your AC

---

### Official Review · Reviewer_xTBx · 2025-10-26

**Soundness:** 4
**Presentation:** 3
**Contribution:** 4
**Rating:** 8
**Confidence:** 4

**Summary:**

This paper aims to disrupt the downstream performance of LVLMs. Through a theoretical analysis of feasibility, VEAttack generates adversarial examples that significantly degrade multiple tasks while achieving notable computational efficiency over other attack approaches.

**Strengths:**

The motivation is clear, and the introduction effectively conveys the idea. VEAttack provides a solid and effective paradigm for gray-box adversarial attacks on LVLMs, offering a detailed analysis and feasibility assessment for this approach. The effectiveness and efficiency are well demonstrated across several datasets and models.

**Weaknesses:**

(1) Table 9 shows the attack performance of the Image-Text Retrieval task, which complements the tasks. However, another focus of these works [1, 2] is on transfer attacks between vision encoders, like ALBEF and CLIP-CNN, and it is recommended to include more demonstrations of this performance.

(2) Eq. (5) gives two baselines, but seems to lack the comparison of the second L2 Attack.

(3) Based on observation 4, you perform a time comparison. However, I notice that the used step is 50 instead of 100 in the setting. I suggest adding a time and effect comparison under the condition of complete alignment.

(4) There is a typo: “SRA” should be “SGA” in Table 9 following [1].

[1] Lu D, Wang Z, Wang T, et al. Set-level guidance attack: Boosting adversarial transferability of vision-language pre-training models[C]//Proceedings of the IEEE/CVF International Conference on Computer Vision. 2023: 102-111.

[2] Gao S, Jia X, Ren X, et al. Boosting transferability in vision-language attacks via diversification along the intersection region of adversarial trajectory[C]//European Conference on Computer Vision. 2024: 442-460.

**Questions:**

(1) The results and trends in Figure 2 (b) are inconsistent with those in the Experiments. How are they obtained?

(2) Figure 7 shows that VEAttack is effective even when the attack step is 30 or even 10. Is this different from other attacks, or do other attacks have the same characteristics?

---

> ### Author Response · Authors · 2025-11-21
> **Response to Reviewer xTBx – Part 1 of 2**
>
> **Dear Reviewer xTBx,**
>
> We greatly appreciate your recognition of this work. Our responses to all your concerns are as follows:
>
> **Response to Weakness 1: Transfer Attacks on Image-Text Retrieval Task.**
>
> Thanks for this nice concern. In the revised manuscript, we have added a new experiment in Table 11 in the revised paper, and also shown in Table 1 below.
>
> We used CLIP-ViT as the source model to attack ALBEF and CLIP-CNN on the Image-Text Retrieval task. While SGA [1] and DRA [2] are specifically engineered with complex augmentation loops to boost transferability, VEAttack achieves competitive results with a much simpler, vision-encoder-only optimization, consuming significantly less memory. A key novelty of our work is the discovery that robust vision encoders yield stronger transfer attacks. To demonstrate this, we simply switched the source model to FARE. This minimal change resulted in a dramatic performance boost, effectively rivaling the performance of the DRA method while maintaining a lower memory footprint.
>
> **Table 1**: Comparison of transfer attack with SGA and DRA on the Image-Text Retrieval task.
> |Method|Source|ALBEF(TR R@1)|ALBEF(IR R@1)|CLIP-CNN(TR R@1)|CLIP-CNN(IR R@1)|Memory|
> |-|-|-|-|-|-|-|
> |SGA|CLIP-ViT|22.42|34.59|53.26|61.1|16.63GB|
> |DRA|CLIP-ViT|27.84|42.84|**64.88**|**69.50**|16.71GB|
> |VEAttack|CLIP-ViT|16.58|32.23|44.06|53.55|**6.75GB**|
> ||FARE|**35.35**|**48.78**|55.17|63.05|10.52GB|
>
> **Response to Weakness 2: Comparison with L2 Attack.**
>
> Thanks for this nice concern. In fact, the L2 Attack represents a naive adversarial attack baseline derived from the adversarial training method FARE [3]. In our original submission, we had included a comparison with this method on image captioning tasks (COCO and Flickr30k) in Table 8. To provide a more comprehensive evaluation as per your suggestion, we have extended this comparison to VQA tasks. As shown in Table 2, VEAttack consistently outperforms the L2 Attack across all four datasets, confirming that attacking patch tokens is more effective than the naive approach of targeting the class token.
>
> **Table 2**: Performance comparison between L2 Attack and VEAttack across Captioning and VQA tasks on LLaVa1.5-7B.
> | Method | COCO | Flickr30k | TextVQA | VQAv2 |
> |--|--|--|--|--|
> | L2 Attack | 25.2 | 20.6 | 17.5 | 49.8 |
> | VEAttack | **7.1** | **6.3** | **10.1** | **38.4** |
>
> **Response to Weakness 3: Time Comparison.**
>
> Thank you for the valuable and insightful comments. We appreciate your attention to detail regarding the experimental settings. We would like to clarify that the time and effect comparison under the condition of $t=100$ was indeed presented in Figure 2 of the original submission. Following your constructive suggestion, we have updated Table 3 in the revised manuscript to strictly distinguish and emphasize the settings. The results confirm that VEAttack achieves highly efficient and effective attacks under both standard and optimized settings, consistently outperforming white-box and gray-box attacks in terms of the trade-off between time consumption and performance degradation.
>
> **Table 3**: Comparison of the effectiveness and efficiency between VEAttack and other white-box and gray-box attacks.
> |Attack|Version|Flops|COCO|Time(h)|Flickr30k|Time(h)|TextVQA|Time(h)|
> |-|-|-|-|-|-|-|-|-|
> |clean|None|99.3G|115.5|1.33|77.5|0.3|37.1|0.42|
> |APGD|White-box|9.93T|13.1|25.5|9.5|5.2|8.1|22.7|
> |Ensemble|White-box|9.93T|**3.1**|41.9|**1.2**|14.3|**0.0**|37.8|
> |VT-Attack|Gray-box|3.04T|12.2|65.0|12.3|13.3|10.0|64.8|
> |VEAttack($\epsilon$=4/255,t=100)|Gray-box|2.59T|7.1|8.5|6.3|1.7|10.1|6.7|
> |VEAttack($\epsilon$=8/255,t=50)|Gray-box|2.59T|5.5|**5.3**|4.5|**1.1**|9.4|**4.1**|
>
> **Response to Weakness 4: Typo Correction.**
>
> We sincerely thank the reviewer for the careful review and for pointing out this typo. We have corrected "SRA" to "SGA" in Table 10 of the revised manuscript.

---

> > ### Author Response · Authors · 2025-11-21
> > **Response to Reviewer xTBx – Part 2 of 2**
> >
> > **Response to Question 1: Consistency of results.**
> >
> > We sincerely apologize for the confusion caused. We would like to clarify that in Figure 2(b), we visualized the percentage of performance drop relative to the clean performance to uniformly illustrate the attack effectiveness across different tasks. To ensure clarity and prevent further misunderstanding, we have updated the y-axis label of Figure 2(b) to "Score Degradation Rate" in the revised manuscript. We thank the reviewer for pointing this out.
> >
> >
> > **Response to Question 2: Lower Attack Steps.**
> >
> > We greatly appreciate the opportunity to clarify this point and provide additional insights based on your valuable feedback. We conducted additional tests on the sensitivity of attack steps using the L2 attack and MIX.Attack, in addition to VEAttack under $\epsilon=8/255$ in Table 4 below. The results indicate that the low sensitivity to attack steps observed in Observation 4 varies across different attack types. Specifically, we found that all attack methods exhibit greater sensitivity to attack steps on the COCO Caption task compared to the VQA task. However, VEAttack demonstrates superior effectiveness, maintaining its performance even with reduced steps. In contrast, methods like MIX.Attack shows a more pronounced decline in effectiveness when steps are lowered significantly.
> >
> > **Table 4**: Ablation of attack steps on different attacks and tasks.
> > **Table:** Ablation of attack steps on different attacks and tasks.
> > | Attack | Task | Steps (10) | Steps (30) | Steps (50) | Steps (80) | Steps (100) |
> > |--|--|--|--|--|--|--|
> > | L2 Attack | COCO | 48.5 | 31.1 | 26.5 | 22.1 | 19.9 |
> > |  | TextVQA  | 19.9 | 15.6 | 14 | 13.2 | 13.3 |
> > | MIX.Attack | COCO | 82.9 | 68.4 | 68.2 | 63.8 | 63.9 |
> > |  | TextVQA  | 25.1 | 21   | 20.6 | 17.4 | 17.9 |
> > | VEAttack | COCO | **17.9** | **8.7** | **5.5** | **6.2** | **5.2** |
> > |  | TextVQA | **17.5** | **11.2** | **9.4** | **8.1** | **7.9** |
> >
> > We hope our responses address your concerns and look forward to your reply and discussion.
> >
> >
> > **References**
> >
> > [1] Lu D, Wang Z, Wang T, et al. Set-level guidance attack: Boosting adversarial transferability of vision-language pre-training models[C]//Proceedings of the IEEE/CVF International Conference on Computer Vision. 2023: 102-111.
> >
> > [2] Gao S, Jia X, Ren X, et al. Boosting transferability in vision-language attacks via diversification along the intersection region of adversarial trajectory[C]//European Conference on Computer Vision. 2024: 442-460.
> >
> > [3] Schlarmann C, Singh N D, Croce F, et al. Robust clip: Unsupervised adversarial fine-tuning of vision embeddings for robust large vision-language models[J]. arXiv preprint arXiv:2402.12336, 2024.

---

> ### Comment · Area_Chair_gUSm · 2025-11-25
>
> Dear Reviewer xTBx,
>
> The authors have responded to your reviews. Please review and provide your feedback and responses.
>
> Best,
>
> Your AC

---

### Official Review · Reviewer_XkoF · 2025-10-31

**Soundness:** 3
**Presentation:** 3
**Contribution:** 3
**Rating:** 6
**Confidence:** 5

**Summary:**

VEAttack presents a gray-box adversarial attack targeting only the vision encoder of large vision-language models (LVLMs).
It minimizes cosine similarity between clean and perturbed patch token features, bypassing the need for task labels or prompt access.
This results in downstream-agnostic perturbations that break captioning, VQA, and hallucination benchmarks simultaneously—while remaining efficient and imperceptible.
The paper provides theoretical grounding showing that patch-level perturbations propagate more strongly to the LLM via alignment layers than class-token perturbations.
Extensive experiments show massive performance drops (up to ~95% in captioning, ~75% in VQA) and strong transferability across models and tasks, far exceeding existing white-box and gray-box baselines.

**Strengths:**

1. Realistic threat model: Attacks only the shared vision encoder—a genuinely deployable setting for LVLM vulnerabilities.
2. Theoretically principled: Clear justification that perturbing patch tokens yields stronger downstream disruption than class tokens.
3. Highly transferable: Single perturbation damages multiple tasks (captioning, VQA, hallucination).
4. Efficiency: 8–13× faster than prior multi-step attacks, with small ε (2–8/255).
5. Insightful analysis: Reveals internal LLM distortions, attention asymmetries (image vs. instruction), and the “Möbius band” paradox—robust encoders yield more transferable attacks.

**Weaknesses:**

1. Defense gap: No practical mitigation or robust-training strategy is explored beyond noting cost trade-offs.
2. Limited architecture diversity: Focuses mainly on CLIP-based encoders; broader evaluation would strengthen claims.
3. Transfer paradox underexplained: The Möbius effect is intriguing but remains a descriptive observation, not a mechanistic analysis.
4. Ethical discussion minimal: Needs clearer guidance on responsible release and safety implications.
5. The paper closely overlaps with the recently released work arXiv:2412.08108, which also investigates adversarial attacks on vision encoders of LVLMs and demonstrates downstream task-agnostic degradation across captioning and VQA. While the two studies share a very similar motivation and methodological framing, the current submission does not cite or discuss this concurrent work.

**Questions:**

Please refer to the Weaknesses section.

---

> ### Author Response · Authors · 2025-11-21
> **Response to Reviewer XkoF – Part 1 of 2**
>
> **Dear Reviewer XkoF,**
>
> We greatly appreciate your recognition of this work. Our responses to all your concerns are as follows:
>
>
> **Response to Weakness 1: Robust-training Strategy.**
>
> Thanks for this nice concern. We agree that exploring robust-training strategies is important. The most straightforward defense method here is deploying the adversarially robust vision encoder. We would like to clarify that we have indeed explored and evaluated these kinds of defense strategies against VEAttack within our paper, specifically in Table 2 of our paper, and also shown in Table 1 below.
>
> **(1) Evaluation of Existing Defenses.** We evaluated our attack against LVLMs equipped with robust vision encoders, specifically TeCoA [1] and FARE [2], which are fine-tuned using Adversarial Training (AT). The results demonstrate that defense mechanisms focusing on the vision encoder are effective. This proves that Adversarial Training on the vision encoder is a reasonable and valid means to counter VEAttack.
>
> **(2) Call for Future Defense Research.** While we validated that AT provides partial mitigation, the performance degradation remains significant. Based on our findings of Möbius Band, we call upon the community to pay greater attention to the defense of the vision encoder, which involves enhancing robustness and developing mechanisms to reduce the transferability of adversarial examples generated from these robust encoders.
>
> **Table 1**: The performance of VEAttack against defense mechanisms, which are TeCoA and FARE.
> |Source\Target|LLaVa-CLIP|LLaVa-TeCoA|LLaVa-FARE|OpenFlamingo-CLIP|OpenFlamingo-TeCoA|OpenFlamingo-FARE|
> |-|-|-|-|-|-|-|
> |clean|115.5|88.3|102.4|79.7|66.9|74.1|
> |CLIP|**3.7**|92.2|100.4|**3.6**|70.7|78.1|
> |TeCoA|22.1|**25.3**|25.1|17.7|**20.8**|22.6|
> |FARE|38.7|54.5|**48.6**|26.3|42.7|**36.4**|
>
> **Response to Weakness 2: Non-CLIP-based Encoders.**
>
> Thank you for raising this important suggestion to improve the completeness of our work. In the revised manuscript, we have added a new experiment in Section F of our revised paper and the corresponding results in Table 9 in the revised paper. Specifically, we evaluate VEAttack on MiniGPT-4 (based on BLIP-2) and mPLUG-Owl2 (based on a distinct vision backbone, MplugOwl) on the COCO Captioning task, as shown in Table 2 below. To ensure a fair comparison, we followed the settings of VT-Attack as $\epsilon = 8/255$, $\alpha=1/255$, and the iterations $t=100$.
>
> The results demonstrate that VEAttack remains highly effective on these diverse architectures. These additional experiments confirm that VEAttack's effectiveness is not limited to CLIP-based models but generalizes well to other advanced vision encoders, strengthening our claim of downstream-agnostic applicability.
>
> **Table 2**: Attack performance comparison on COCO Captioning against more LVLMs.
> |Models|Attack|BLUE@4|METEOR|ROUGE_L|CIDEr|SPICE|
> |-|-|-|-|-|-|-|
> |MiniGPT-4|Clean|25.0|27.9|53.1|94.0|21.3|
> ||AttackVLM-ii|14.5|20.8|41.9|48.9|13.7|
> ||VT-Attack|10.2|18.7|37.1|24.8|11.2|
> ||VEAttack|**7.8**|**14.9**|**32.2**|**13.7**|**7.1**|
> |mPLUG-Owl2|Clean|38.6|30.2|59.5|139.2|24.3|
> ||AttackVLM-ii|29.1|25.6|52.7|101.1|18.9|
> ||VT-Attack|29.4|25.2|52.7|100.3|19.1|
> ||VEAttack|**18.3**|**20.0**|**44.9**|**65.1**|**13.4**|
>
> **Response to Weakness 3: Mechanistic Explanation of Möbius Effect.**
>
> Thanks for this nice concern. We agree that the Möbius effect warrants a deeper mechanistic explanation beyond a surface-level observation.
>
> In the revised version, we have added a new section (Section J) titled "Mechanistic Analysis on Möbius band," accompanied by a heatmap visualization (Figure 9 in the revised paper).
>
> In this analysis, we track the deviation of patch token values throughout the optimization process, and the visualization reveals the underlying mechanism:
>
> - **Robust-to-Standard.** The heatmap shows intense, deep colors appearing early in the optimization. This indicates that to break a robust encoder like FARE, the attack is forced to generate large-magnitude feature distortions. These strong perturbations result in severe feature collapse when transferred to a standard model, explaining the high transferability.
>
> - **Standard-to-Robust.** The heatmap remains very light. This suggests that perturbations optimized on CLIP are superficial, which may be sufficient to fool CLIP, but are easily filtered out by FARE's robust feature space. The attack on CLIP fails to find a direction that significantly impacts the robust model's representation.
>
> This analysis mechanistically bridges the gap between model robustness and attack transferability, confirming that robust source models compel the generation of universally destructive feature perturbations.

---

> > ### Author Response · Authors · 2025-11-21
> > **Response to Reviewer XkoF – Part 2 of 2**
> >
> > **Response to Weakness 4: Ethical Discussion.**
> >
> > We sincerely thank the reviewer for emphasizing this critical aspect. We fully agree that a work demonstrating effective attacks must be accompanied by a clear and responsible discussion of its safety implications.
> >
> > In the revised manuscript, we have added a dedicated Ethical Statement and Responsible Disclosure in Section A to explicitly address these concerns, which cover two key points:
> >
> > - **Responsible Release.** We clarify that our code and data will be released under a restrictive license strictly for academic and research use. We have also ensured that our experimental design focuses solely on technical performance metrics on standard benign datasets, avoiding any generation of harmful, toxic, or sensitive content.
> >
> > - **Guidance for Future Defense.** We explicitly discuss the broader safety implications of our "Möbius Band" observation. By revealing that robust encoders can facilitate stronger transfer attacks, we highlight a systemic risk in current defense paradigms that rely solely on adversarial training. Based on it, we provide actionable guidance for the community, calling for a shift towards "transfer-resistant" encoder design and multi-stage verification pipelines.
> >
> > We believe this expanded discussion aligns with the practices of the adversarial machine learning community and sufficiently addresses the ethical dimensions of our work.
> >
> > **Response to Weakness 5: Discussion of Concurrent Work.**
> >
> > Thank you for providing this concurrent work (arXiv:2412.08108) and constructive suggestions. We have analyzed the paper Doubly-UAP [3] you provided in comparison to our VEAttack. Below is a detailed explanation and comparison:
> >
> > Doubly-UAP [3] makes a significant contribution by introducing the first Universal Adversarial Perturbation specifically designed for VLMs. It achieves an impressive doubly universal effect across both images and text prompts, offering a robust label-free optimization strategy. However, our VEAttack differs fundamentally from Doubly-UAP in terms of Attack Paradigm, Optimization Objectives, and Unique Insights:
> >
> > - **Attack Paradigm.** The goal of Doubly-UAP is to train a single, fixed perturbation that works across any image. This comes with a training cost, which requires a dataset of 200,000 images (from ImageNet) and 3 training epochs. In contrast, VEAttack is an image-specific attack, which eliminates the need for any pre-training, large-scale datasets, or extensive offline computation.
> > - **Optimization Objectives.** Doubly-UAP targets Attention Layers in the middle-to-late layers of the vision encoder. VEAttack targets Output Representations, which do not need to access or analyze internal layers.
> > - **Unique Insights.** We reveal the paradox where robust encoders facilitate stronger transfer attacks in the "Möbius Band" effect.
> >
> > We have updated our "Related Work" section to cite and discuss this excellent work [3], explicitly clarifying these distinctions to ensure our novelty is well-defined.
> >
> > We hope our responses address your concerns and look forward to your reply and discussion.
> >
> >
> > **References**
> >
> > [1] Mao C, Geng S, Yang J, et al. Understanding zero-shot adversarial robustness for large-scale models[J]. arXiv preprint arXiv:2212.07016, 2022.
> >
> > [2] Schlarmann C, Singh N D, Croce F, et al. Robust clip: Unsupervised adversarial fine-tuning of vision embeddings for robust large vision-language models[J]. arXiv preprint arXiv:2402.12336, 2024.
> >
> > [3] Kim H S, Kim M, Kim C. Doubly-universal adversarial perturbations: Deceiving vision-language models across both images and text with a single perturbation[J]. arXiv preprint arXiv:2412.08108, 2024.

---

> ### Comment · Area_Chair_gUSm · 2025-11-25
>
> Dear Reviewer XkoF,
>
> The authors have responded to your reviews. Please review and provide your feedback and responses.
>
> Best,
>
> Your AC

---

### Official Review · Reviewer_UHud · 2025-10-31

**Soundness:** 2
**Presentation:** 2
**Contribution:** 2
**Rating:** 2
**Confidence:** 4

**Summary:**

This paper focuses on addressing the vulnerability of Large Vision-Language Models (LVLMs) to adversarial attacks and proposes a novel gray-box attack method called VEAttack. Unlike existing white-box attacks that require full-model gradients and task-specific labels (resulting in high costs scaling with tasks) and black-box attacks that depend on proxy models (needing large perturbation sizes), VEAttack targets only the vision encoder of LVLMs.

**Strengths:**

1. Innovative Attack Setting: focus on the vision encoder in a gray-box setting.

**Weaknesses:**

1. Lack of citations and comparisons with papers highly similar to this paper.

- An Image Is Worth 1000 Lies: Adversarial Transferability across Prompts on Vision-Language Models, ICLR 2024.
- QAVA: Query-Agnostic Visual Attack to Large Vision-Language Models, NAACL 2025.
- InstructTA: Instruction-Tuned Targeted Attack for Large Vision-Language Models, ARXIV.

2. Without Any New Insights: Attacking the vision encoder to achieve attacks on the entire LVLMs is not novel; it is quite intuitive.

3. Sensitivity to Vision Encoder Type: VEAttack’s effectiveness heavily relies on the vision encoder of LVLMs. For LVLMs using non-CLIP vision encoders, the paper’s experiments show relatively limited attack effects. Can this be called a gray-box attack?

4. Limited Transferability from Large-Scale Vision Encoders: Experimental results show that when using the vision encoders of large models (e.g., mPLUG-Owl2, Qwen-VL) as source models for transfer attacks, it is difficult to achieve successful attacks on other models. The paper only provides a preliminary empirical explanation but lacks in-depth analysis of the underlying reasons (e.g., differences in feature representation mechanisms between large and small models).

5. Narrow Scope of Evaluation Tasks: While the paper evaluates VEAttack on image captioning, VQA, and hallucination benchmarks, it does not test its performance on other important LVLM tasks such as image-text retrieval (only a brief ASR evaluation is provided) or visual grounding, which limits the demonstration of its generalizability.

6. Overstatement on Imperceptibility: The paper claims that VEAttack has high imperceptibility through visual inspection of perturbation images, in fact, all adversarial examples satisfy this. This makes absolutely no sense.

7. Lack of Defense Mechanism Research: The paper only proposes the VEAttack method but does not explore corresponding defense strategies to counter it.

**Questions:**

See weaknesses.

---

> ### Author Response · Authors · 2025-11-21
> **Response to Reviewer UHud – Part 1 of 4**
>
> **Dear Reviewer UHud,**
>
> Thanks for the constructive comments, which enabled us to improve this paper. We hope our rebuttal could adequately address your concerns as follows:
>
> **Response to Weakness 1: Comparisons with relevant papers.**
>
> Thank you for providing additional references and constructive suggestions. We provide a detailed discussion of these works and the compared results to our VEAttack if the setting is feasible for comparison. Below is a detailed discussion and comparison:
>
> **(1) Compare with CroPA [1]**
>
> CroPA makes a pioneering contribution by defining "cross-prompt adversarial transferability" and proposing learnable prompts to mislead VLMs across diverse textual inputs. Compared with CroPA, our VEAttack targets a different gray-box scenario, prioritizing efficiency and architecture independence. The detailed distinctions are as follows:
>
> - **Distinct Threat Models.** CroPA requires full access to the entire LVLM, including the Large Language Model (LLM) parameters, to compute the language modeling loss (Eq. 1 and Eq. 2 in [1]). In contrast, our VEAttack operates under a strictly constrained setting, where the attacker has only access to the Vision Encoder. We do not access the LLM, the projection layer, or the textual output probabilities.
>
> - **Different Optimization Objectives.** CroPA optimizes image perturbations by leveraging learnable prompts, and it focuses on transferability by covering the text embedding space. However, our VEAttack optimizes perturbations solely in the visual feature space.
>
> - **Efficiency.** Since CroPA requires calculating gradients through the massive LLM, it incurs significant computational/memory costs. VEAttack, targeting only the standalone vision encoder, is more efficient. As shown in Table 3, VEAttack is 8 times faster than ensemble methods involving LLMs.
>
> **(2) Compare with QAVA [2]**
>
> QAVA offers significant insights into the vulnerability of visual-language alignment modules (e.g., Q-Former), effectively proposing a strategy to generate robust adversarial examples against unknown questions by leveraging randomized text queries. Compared with QAVA, VEAttack offers a more generalized solution for modern LVLMs. Detailed comparisons are as follows:
>
> - **Architecture-Specific.** QAVA is designed for models with a query-based alignment module (e.g., Q-Former). This makes it difficult to apply directly to models like LLaVA, which use a simple linear projector. In contrast, VEAttack targets the Vision Encoder directly and is universally applicable in a gray-box setting across all these architectures without modification. As shown in Table 1, VEAttack provides a higher attack performance, highlighting the generalization of our methods.
> - **Multi-modal Optimization.** QAVA requires sampling and processing a batch of random text questions to optimize the adversarial image. In contrast, VEAttack is strictly vision-only, requiring no text inputs during attack generation. This reduces computational overhead and eliminates the need to construct surrogate question datasets.
>
> **Table 1**: Comparison of the attack performance (↓) on LLaVA1.5-7B and the COCO captioning task with different evaluation metrics.
>
> |Method|BLUE@4|METEOR|ROUGE_L|CIDEr|SPICE|
> |-|-|-|-|-|-|
> |QAVA|22.3|24.7|49.2|79.9|17.8|
> |VEAttack|4.4|10.9|30.7|7.1|2.9|
>
> **(3) Compare with InstructTA [3]**
>
> InstructTA proposes a novel gray-box targeted attack framework that cleverly leverages GPT-4 and text-to-image models to infer instructions and generate target images, enabling effective targeted attacks without accessing the LLM. However, VEAttack is distinct from InstructTA in terms of adversarial objectives, resource dependencies, and architectural generality:
>
> - **Surrogate Model Composition.** Although InstructTA avoids accessing the LLM, its surrogate model $M$ is explicitly defined as comprising the visual encoder and a pre-trained projector (e.g., Q-Former). In contrast, VEAttack operates with a minimal constraint, targeting the Vision Encoder alone.
> - **Adversarial Objectives.** InstructTA requires a specific target text and utilizes a text-to-image model (e.g., Stable Diffusion) to generate a target image. In contrast, we do not require any target text, nor do we rely on external generative models. Our objective is to disrupt the semantic representation of the vision encoder. Since the official implementation code for InstructTA was not publicly available, we were unable to conduct a direct quantitative performance comparison.
>
> We sincerely thank the reviewer for identifying these relevant papers. We have updated the "Related Work" section and Table 8 in our revised manuscript to explicitly cite and discuss CroPA [1], QAVA [2], and InstructTA [3].

---

> > ### Author Response · Authors · 2025-11-21
> > **Response to Reviewer UHud – Part 2 of 4**
> >
> > **Response to Weakness 2: Clarifying the Novelty and Insights of VEAttack.**
> >
> > We agree that targeting the vision encoder is an intuitive solution to enhance attack generalization, given its pivotal role in LVLMs, which is also adopted by some previous works. However, VEAttack distinguishes itself by bridging the gap between this intuition and a rigorous as well as effective methodology through the following contributions:
> >
> > **(1) Bridging the gap between a naive idea and a theoretically grounded method.** While the concept is intuitive, effectively executing it without accessing the LLM or text modality is non-trivial. We provide the foundational theoretical derivation (Proposition 1) to establish the feasibility of vision-only attacks, proving a lower bound for perturbation propagation to aligned features2. Furthermore, we mathematically prove and empirically validate (Proposition 2) that the "standard" approach of perturbing the class token ($z_{cls}$) is insufficient compared to patch tokens ($z_v$). This transforms a general intuition into a concrete, mathematically justified framework.
> >
> > **(2) Superior Gray-box Performance.** We focus on exploring how to fully exploit the attacking capability of vision encoders using only the vision encoder and image modality information. And we conducted extensive performance and efficiency testing on current gray-box attacks. VEAttack demonstrates that utilizing solely the vision encoder and image modality achieves superior generalization and efficiency (8 times faster than white-box ensemble attacks).
> >
> > **(3) Unique Insights on Transferability.** Our work uncovers distinct phenomena that provide fresh value to the community:
> >
> > - **The "Möbius Band" Effect (Observation 3).** We reveal a paradox where robust vision encoders (like FARE) improve model defense, yet adversarial examples generated against them exhibit stronger transferability to other models.
> > - **Transferability of Large Encoders.** In Table 10, we investigate the transferability of large-scale specific vision encoders. We find that transferring attacks from these specialized large encoders is challenging due to feature specialization, providing crucial empirical guidance for selecting source models in transfer attacks.
> > - We have included a deeper discussion of these unique insights in Sections I and J of our revised manuscript to further highlight the novelty of our contributions.
> >
> > **Response to Weakness 3: Sensitivity to Vision Encoder Type.**
> >
> > Thanks for this nice concern. We respectfully point out that this concern may stem from a misunderstanding of the distinction between our Gray-box (direct) and Black-box (transfer) experimental settings. We firmly maintain that VEAttack constitutes a valid and highly effective Gray-box attack across diverse architectures.
> >
> > **(1) Clarification of the "Gray-box" Definition.** We strictly follow the standard definition of a gray-box attack, which assumes the attacker has access to partial model components, which are the parameters of the vision encoder $\theta_V$. Therefore, when attacking an LVLM that uses a non-CLIP encoder (e.g., mPLUG-Owl2), the gray-box setting implies the attacker utilizes the gradients of that specific non-CLIP encoder, MplugOwl.
> >
> > **(2) Clarification of the Performance on Non-CLIP Models.** Table 2 demonstrates a Black-box Transfer setting, where we use CLIP as a source to attack disparate targets like mPLUG-Owl2. The purpose of Table 2 is to illustrate Observation 3 (the "Möbius band" insight): that robust encoders (like FARE) transfer better than standard CLIP even in challenging black-box scenarios. The gray-box performance for non-CLIP models is presented in Table 13 (Appendix I) of the revised paper and also shown in Table 2 of the response, where the source and target encoders match.
> > - mPLUG-Owl2: When directly attacking its vision encoder (Gray-box), the CIDEr score drops drastically from 132.3 to 20.5.
> > - Qwen-VL: When directly attacking its vision encoder (Gray-box), the CIDEr score drops drastically from 138.5 to 27.8.

---

> > > ### Author Response · Authors · 2025-11-21
> > > **Response to Reviewer UHud – Part 3 of 4**
> > >
> > > **Response to Weakness 4: Analysis of Transferability from Large-Scale Vision Encoders.**
> > >
> > > Thank you very much for your constructive comments, which enabled us to polish the paper. We have addressed this concern through both additional empirical verification and in-depth feature analysis.
> > >
> > > **(1) Verification on VT-Attack.** First, to verify whether this limitation is a general issue of existing attacking algorithms, we extended our experiments to include VT-Attack in the same transfer setting. In Table 2, we observed that VT-Attack also struggles to transfer effectively from these large-scale specific vision encoders to other models, which suggests that the difficulty in transferring from specialized large encoders is a general issue.
> > >
> > > **(2) Feature Representation Analysis.** To uncover the underlying reasons, in Section I of the revised version, we have added a feature representation analysis accompanied by a t-SNE visualization in Figure 8 of our revised paper. This analysis proves that the feature spaces of large-scale specific encoders (e.g., mPLUG-Owl2, Qwen-VL) are distinct and isolated from one another. This feature shift creates a structural barrier for transferability, confirming that perturbations optimized for one specific high-dimensional manifold do not align with others. We believe this observation sheds new light on the trade-off between model specialization and adversarial transferability, and we will conduct a more comprehensive investigation into this interesting phenomenon in our future work.
> > >
> > > **Table 2**: Transfer attack on COCO captioning from mPLUG-Owl2 and Qwen-VL to other models.
> > > |Method|Source|Target|CIDEr|Source|Target|CIDEr|
> > > |-|-|-|-|-|-|-|
> > > |VEAttack|mPLUG-Owl2|mPLUG-Owl2|**20.5**|Qwen-VL|Qwen-VL|**27.8**|
> > > |||Qwen-VL|131.7||mPLUG-Owl2|127.1|
> > > |||LLaVa1.5-7B|131.5||LLaVa1.5-7B|138.5|
> > > |||OpenFlamingo-9B|108.2||OpenFlamingo-9B|102.8|
> > > |VT-Attack|mPLUG-Owl2|mPLUG-Owl2|63.3|Qwen-VL|Qwen-VL|60.8|
> > > |||Qwen-VL|134.5||mPLUG-Owl2|130.5|
> > > |||LLaVa1.5-7B|139.0||LLaVa1.5-7B|138.8|
> > > |||OpenFlamingo-9B|100.5||OpenFlamingo-9B|100.3|
> > >
> > > **Response to Weakness 5: More Evaluation Tasks.**
> > >
> > > We sincerely thank the reviewer for this constructive suggestion. First, we would like to clarify that our work follows the standard adversarial attacks on LVLMs [1, 2, 3], which typically validate generalizability across Captioning, VQA, and Classification. Thus, we have extensively validated VEAttack across 5 distinct tasks and 18 datasets:
> > > - Image Captioning: COCO, Flickr30k
> > > - VQA: TextVQA, VQAv2, OK_VQA
> > > - Hallucination: POPE
> > > - Classification: 12 datasets, including Tiny-ImageNet, CIFAR-10/100, ImageNet, etc. (as detailed in Section K).
> > > - Image-Text Retrieval: Flickr30k.
> > >
> > > But we agree that more evaluation on diverse tasks could provide more solid evidence. To further address your concern, we have added experiments on the Visual Grounding task. We evaluated VEAttack against ReCLIP[4], which is a popular zero-shot visual grounding baseline on the RefCOCO and RefCOCO+ datasets with $\epsilon=2/255$ and $t=100$ iterations.
> > >
> > > As shown in Table 3 (and Table 12 in the revised manuscript), VEAttack achieves significant performance degradation compared to baselines, proving its effectiveness on spatial localization tasks. We have updated the revised paper to include these results.
> > >
> > > **Table 3**: Attack performance after different methods on the Visual Grounding task.
> > > |Method|RefCOCO-TestA|RefCOCO-TestB|RefCOCO-Val|RefCOCO+-TestA|RefCOCO+-TestB|RefCOCO+-Val|
> > > |-|-|-|-|-|-|-|
> > > |Clean|49.0|53.9|48.2|50.8|49.1|47.3|
> > > |AttackVLM-ii|27.6|33.5|28.9|28.5|**27.9**|29.1|
> > > |VT-Attack|35.4|40.7|34.5|34.7|37.9|35.7|
> > > |VEAttack|**23.1**|**32.1**|**27.1**|**24.6**|29.6|**27.4**|

---

> ### Author Response · Authors · 2025-11-21
> **Response to Reviewer UHud – Part 4 of 4**
>
> **Response to Weakness 6: Explanation of Imperceptibility.**
>
> Thanks for this nice concern. We acknowledge that in the ideal definition of adversarial examples, perturbations should be imperceptible. However, in the context of black-box attacks on LVLMs, existing methods often require large perturbation budgets, e.g., $\epsilon=16/255$ in [5], or introduce semantical noise to generate adversarial samples. Our claim of "High Imperceptibility" highlights that VEAttack achieves effective disruption using smaller and less structured perturbations compared to black-box methods. To clarify this, we have added a new section L.2 with visual comparisons and quantitative metrics in the revised paper. In Figure 11 of the revised paper, black-box methods like M-Attack often introduce visible texture distortion. In contrast, VEAttack remains visually clean. We computed statistics on 500 sampled adversarial images. VEAttack maintains an average $L_2$ distance of 4.39, whereas M-Attack requires a distance of 15.34. Despite the smaller visual footprint, VEAttack achieves a lower CLIP feature similarity score compared to the clean image, demonstrating that it induces a larger deviation in the semantic feature space with significantly less visual cost. We have updated the manuscript to frame our imperceptibility claim within this comparative context explicitly.
>
> **Response to Weakness 7: Defense Mechanism Research.**
>
> Thank you for raising this important question. We agree that exploring robust-training strategies is important. The most straightforward defense method here is deploying the adversarially robust vision encoder. We would like to clarify that we have indeed explored and evaluated these kinds of defense strategies against VEAttack within our paper, specifically in Table 2 of our paper, and also shown in Table 4 below.
>
> **(1) Evaluation of Existing Defenses.** We evaluated our attack against LVLMs equipped with robust vision encoders, specifically TeCoA [6] and FARE [7], which are fine-tuned using Adversarial Training (AT). The results demonstrate that defense mechanisms focusing on the vision encoder are effective. This proves that Adversarial Training on the vision encoder is a reasonable and valid means to counter VEAttack.
>
> **(2) Call for Future Defense Research.** While we validated that AT provides partial mitigation, the performance degradation remains significant. Based on our findings of Möbius Band, we call upon the community to pay greater attention to the defense of the vision encoder, which involves enhancing robustness and developing mechanisms to reduce the transferability of adversarial examples generated from these robust encoders.
>
> **Table 4**: The performance of VEAttack against defense mechanisms, which are TeCoA and FARE.
> |Source\Target|LLaVa-CLIP|LLaVa-TeCoA|LLaVa-FARE|OpenFlamingo-CLIP|OpenFlamingo-TeCoA|OpenFlamingo-FARE|
> |-|-|-|-|-|-|-|
> |clean|115.5|88.3|102.4|79.7|66.9|74.1|
> |CLIP|**3.7**|92.2|100.4|**3.6**|70.7|78.1|
> |TeCoA|22.1|**25.3**|25.1|17.7|**20.8**|22.6|
> |FARE|38.7|54.5|**48.6**|26.3|42.7|**36.4**|
>
> We hope our responses address your concerns and look forward to your reply and discussion.
>
> **References**
>
> [1] Luo H, Gu J, Liu F, et al. An image is worth 1000 lies: Adversarial transferability across prompts on vision-language models[J]. arXiv preprint arXiv:2403.09766, 2024.
>
> [2] Zhang Y, Xie R, Chen J, et al. Qava: Query-agnostic visual attack to large vision-language models[C]. arXiv preprint arXiv:2504.11038, 2025.
>
> [3] Wang X, Ji Z, Ma P, et al. Instructta: Instruction-tuned targeted attack for large vision-language models[J]. arXiv preprint arXiv:2312.01886, 2023.
>
> [4] Subramanian S, Merrill W, Darrell T, et al. Reclip: A strong zero-shot baseline for referring expression comprehension[J]. arXiv preprint arXiv:2204.05991, 2022.
>
> [5] Li Z, Zhao X, Wu D D, et al. A frustratingly simple yet highly effective attack baseline: Over 90% success rate against the strong black-box models of gpt-4.5/4o/o1[J]. arXiv preprint arXiv:2503.10635, 2025.
>
> [6] Mao C, Geng S, Yang J, et al. Understanding zero-shot adversarial robustness for large-scale models[J]. arXiv preprint arXiv:2212.07016, 2022.
>
> [7] Schlarmann C, Singh N D, Croce F, et al. Robust clip: Unsupervised adversarial fine-tuning of vision embeddings for robust large vision-language models[J]. arXiv preprint arXiv:2402.12336, 2024.

---

> ### Comment · Area_Chair_gUSm · 2025-11-25
>
> Dear Reviewer UHud,
>
> The authors have responded to your reviews. Please review and provide your feedback and responses.
>
> Best,
>
> Your AC

---

### Author Response · Authors · 2025-11-24
**Response to All**

**Dear reviewers and area chairs,**

We sincerely thank all reviewers and area chairs for their time, meticulous review, and constructive comments, which have been instrumental in refining the quality and scope of our paper. We are especially encouraged by the recognition of our work's value:

- Reviewer UHud acknowledged the "**Innovative Attack Setting**," recognizing our focus on the vision encoder in a gray-box setting as a strength.

- Reviewer XkoF appreciated the "**realistic threat model**" and "**theoretically principled**" approach, noting that our method is "**highly transferable**" and offers "**insightful analysis**" on internal LLM distortions.

- Reviewer xTBx highlighted that VEAttack "**provides a solid and effective paradigm for gray-box adversarial attacks**" and that our "**effectiveness and efficiency are well demonstrated**."

- Reviewer eK4n commended the paper for being "**clearly structured and articulately written**" with "**detailed analysis and keen observation**," and acknowledged the "**varied set of datasets and models**" in our experiments.

We have carefully addressed all reviewer comments and significantly revised the manuscript. The major improvements and additions in the revised version are summarized below:

1. Expanded Evaluation on Diverse Architectures and Tasks:

- Following the suggestions of Reviewer UHud and Reviewer XkoF, we have extended our evaluation to include non-CLIP vision encoders. We added experiments on MiniGPT-4 and mPLUG-Owl2 in **Section F (Table 9)**, demonstrating that VEAttack generalizes effectively to diverse architectures beyond CLIP.

- Following the suggestions of Reviewer UHud, we have broadened the task scope by adding the Visual Grounding task. We evaluated VEAttack against ReCLIP on RefCOCO and RefCOCO+ datasets in **Section H (Table 12)**, confirming its effectiveness in spatial reasoning tasks.

- Following the suggestions of Reviewer xTBx, we added the Image-Text Retrieval task on Flickr30k and conducted a comprehensive transfer attack comparison with methods (SGA and DRA) in **Section G.2 (Table 11)**, highlighting our efficiency advantage.

2. In-depth Mechanistic Analysis and Visualizations:

- Following the suggestions of Reviewer UHud, we have added a t-SNE analysis in **Section I (Figure 8)** to visually explain the limited transferability from large-scale specific vision encoders (e.g., Qwen-VL), attributing it to distinct feature shifts.

- Following the suggestions of Reviewer XkoF, we moved beyond descriptive observations of the "Möbius Band" effect by adding a mechanistic analysis in **Section J (Figure 9)**. The new heatmap visualization tracks token value deviations across attack steps, revealing how robust encoders force the generation of universally destructive features.

- Following the suggestions of Reviewer UHud and Reviewer eK4n, we added **Section L.2 (Figure 11)** to provide both qualitative visualizations and quantitative metrics (L2 distance and CLIP Score) comparing VEAttack with the black-box M-Attack. This objectively substantiates our claim of superior imperceptibility and efficiency.

3. Clarifications on Settings and Ethics:

- Following the suggestions of Reviewer XkoF, we have added a dedicated Ethical Statement in **Section A**, discussing responsible release protocols and safety implications for future defense research.

- Following the suggestions of Reviewer eK4n and Reviewer xTBx, we have clarified the efficiency comparison settings in Table 3 and ensured consistent metric reporting (Score Degradation Rate) in **Figure 2(b)** and **Table 1**.

All modifications in the main text are marked in blue for easy tracking. We hope these revisions and our detailed responses have fully addressed your concerns. We look forward to any further discussion.

We thank all reviewers and area chairs again for your dedication.

Best,

Authors of Paper #363

---

### Meta-Review · Area_Chair_SknW · 2025-12-06

**Summary:**

This paper proposes a transferable attack, VEAttack, against LVLMs. Its key contribution lies in leveraging feature dissimilarity to improve transferability, enabling strong cross-model transfer even under small perturbation budgets. I do not view the grey-box aspect as particularly novel, as many prior works also target the vision encoder and some are downstream-agnostic (in the CLIP- > downstream setting). However, the untargeted attack setting appears to be much less explored, and thus provides genuine novelty. The authors conduct extensive experiments that effectively shed light on the factors influencing transferability, which I consider a valuable contribution. The theoretical analysis further strengthens the work.

Four reviewers initially rated the paper as 8, 8, 6, and 2, yielding an average of 6. The reviewer who gave a rating of 2 raised concerns about novelty, missing experiments, and the absence of an ethics statement; these concerns are legitimate. However, the authors have provided substantial and satisfactory responses, addressing these issues as well as other reviewers’ comments.

Overall, I recommend acceptance of this paper. Despite some weaknesses, they are not fundamental, and the work is solid and contributes meaningfully to the field.

**Reviewer Concerns:**

The authors are encouraged to include a more detailed comparison table summarizing the settings of existing works, including their threat models and perturbation budgets, for both LVLM attacks and CLIP-to-downstream attack methods.

**Reviewer Scores:**

xTBx:8
eK4n: 8
XkoF: 6
UHud: 4

---

### Decision · Program_Chairs · 2026-01-26

Accept (Poster)